# Electrochemical nitrate reduction in acid enables high-efficiency ammonia synthesis and high-voltage pollutes-based fuel cells

Rong Zhang[1], Chuan Li[1], Huilin Cui[1], Yanbo Wang[1], Shaoce Zhang[1], Pei Li[1], Yue Hou[1], Ying Guo [2] ✉, Guojin Liang[1], Zhaodong Huang[1], Chao Peng [3] ✉ & Chunyi Zhi [1,4,5] ✉

Most current research is devoted to electrochemical nitrate reduction reaction for ammonia synthesis under alkaline/neutral media while the investigation of nitrate reduction under acidic conditions is rarely reported. In this work, we demonstrate the potential of $TiO_2$ nanosheet with intrinsically poor hydrogen-evolution activity for selective and rapid nitrate reduction to ammonia under acidic conditions. Hybridized with iron phthalocyanine, the resulting catalyst displays remarkably improved efficiency toward ammonia formation owing to the enhanced nitrate adsorption, suppressed hydrogen evolution and lowered energy barrier for the rate-determining step. Then, an alkaline-acid hybrid Zn-nitrate battery was developed with high open-circuit voltage of 1.99 V and power density of 91.4 mW cm$^{-2}$. Further, the environmental sulfur recovery can be powered by above hybrid battery and the hydrazine-nitrate fuel cell can be developed for simultaneously hydrazine/nitrate conversion and electricity generation. This work demonstrates the attractive potential of acidic nitrate reduction for ammonia electrosynthesis and broadens the field of energy conversion.

Ammonia ($NH_3$) is an essential raw ingredient for fertilizers pharmaceutical, and nitrogen-containing chemicals industries[1–3]. Under the background of "hydrogen energy" and "carbon neutrality", $NH_3$ also becomes an ideal energy and hydrogen carrier and a perspective transportation fuel alternative with a high gravimetric energy density[4,5]. Recently, electrochemical nitrate reduction reaction ($NO_3^-$RR) for $NH_3$ synthesis has received much attentions[6,7]. The $NO_3^-$ has good availability and rich abundance in nature, especially in wastewater. Converting $NO_3^-$ into $NH_3$ is highly attractive for the treatment of $NO_3^-$-containing wastewater and "turn waste into wealth"[8,9]. In addition, $NO_3^-$RR exhibits far superior $NH_3$-yield efficiency and much lower energy consumption than nitrogen reduction and traditional

Haber-Bosch process, respectively, owing to the enhanced adsorption behavior and low activation energy of $NO_3^-$ on the catalyst surface[10–12]. $NO_3^-$RR, therefore, is of great significance for environmental protection, green $NH_3$ production, and energy utilization.

Extensive efforts have been devoted to exploring selective electrocatalysts for the $NO_3^-$RR in alkaline/neutral media[7,13,14]. Although the high Faradic efficiencies (FE) towards $NH_3$ (>90%) have been achieved on state-of-art catalysts, the nitrite ($NO_2^-$) is usually generated as the main byproduct at the beginning and large overpotentials are required to achieve the best $NO_3^-$-to-$NH_3$ performance. It is also suggested that in situ generated $NH_3$ exists in gaseous form at the electrode surface due to the increased local pH (mostly over 10) and may be released

[1]Department of Materials Science and Engineering, City University of Hong Kong, 83 Tat Chee Avenue, 999077 Hong Kong, China. [2]College of Materials Science and Engineering, Shenzhen University, 518061 Shenzhen, China. [3]Multiscale Crystal Materials Research Center, Shenzhen Institute of Advanced Technology, Chinese Academy of Sciences, 518055 Shenzhen, China. [4]Centre for Functional Photonics, City University of Hong Kong, 999077 Kowloon, Hong Kong, China. [5]Songshan Lake Materials Laboratory, 523808 Dongguan, Guangdong, China. ✉e-mail: yingguo@szu.edu.cn; chao.peng@siat.ac.cn; cy.zhi@cityu.edu.hk

from the alkaline/neutral electrolyte, which needs an acid adsorption process for $NH_3$ capture[15,16]. According to the reaction equation of $NO_3^- + 6H_2O + 8e^- \rightarrow NH_3 + 9OH^-$ in alkaline/neutral media, the $NO_3^-$RR involves nine proton-coupled electron transfer, where protons are produced by an additional water dissociation step ($H_2O \rightarrow H^+ + OH^-$)[17–19], which may result in large overpotentials and sluggish kinetics for limited $NO_3^-$RR performance (Fig. 1a). In contrast, direct nitrate reduction under strongly acidic conditions offers unique advantages compared with neutral/alkaline conditions. For example, the volatilization of $NH_3$ in neutral/alkaline electrolytes can be avoided, and ammonium fertilizers/salts (i.e., $NH_4NO_3$ and $(NH_4)_2SO_4$) can be obtained directly to be absorbed by the plants[20,21]. As the reaction becomes more acidic, abundant protons are provided for continuous hydrogenation reactions of $NO_3^-$, which may guarantee an enhanced $NO_3^-$ conversion rate and generate $NH_3$ more energy-efficiently. All these considerations reveal the great potential of acidic $NO_3^-$RR for $NH_3$ production, which, however, is rarely explored now.

Hydrogen evolution reaction (HER) inevitably thus becomes a competitive reaction for $NO_3^-$RR in acidic media[22–24]. Up to now, most reported $NO_3^-$RR electrocatalysts working at pH≥7 are based on late-transition metals (such as Cu, Fe, Co, and Ni)[13,25,26]. Unfortunately, most of them are unstable in acid conditions and few are applied for acidic $NO_3^-$RR[27]. In addition, these electrocatalysts may suffer from the reduced $NH_3$ FE due to the obviously enhanced competitive HER in acidic media. Herein, we reported the application of Fe phthalocyanine/$TiO_2$ (FePc/$TiO_2$) as a stable and active electrocatalyst for energy-efficient $NO_3^-$RR in acid (pH = 1) with impressive $NH_3$ yield rate of 17.4 mg $h^{-1}$ $cm^{-2}$ and a $NH_3$ FE of 90.6%. Such FePc/$TiO_2$ exhibits poor HER activity and enhanced $NO_3^-$ adsorption, fascinating the selective $NO_3^-$RR to $NH_3$. In-situ Fourier transform infrared spectroscopy (FTIR) and theoretical calculation reveals the reaction pathway over FePc/$TiO_2$ with *NO → *NOH as rate-determining step. The developed alkaline-acid hybrid Zn−$NO_3^-$ battery (AAHZNB) based on FePc/$TiO_2$ cathode shows a high open-circuit voltage (OCV) up to 1.99 V with a

high power density of 91.4 mW $cm^{-2}$ for electricity supply and $NH_3$ synthesis, which can be applied for efficiently environmental sulfur recovery. Moreover, the Zn anode in the AAHZNB can be replaced by hydrazine ($N_2H_4$) for additional $N_2H_4$ pollutant removal in a $N_2H_4$-$NO_3^-$ fuel cell.

## Results
$TiO_2$, with poor HER activity and notable corrosion resistance, is an appropriate and versatile electrocatalytic material for $NO_3^-$RR[28]. We first explored the $NO_3^-$RR performance of $TiO_2$ under acidic, neutral, and alkaline conditions. As shown in Fig. 1b, the linear sweep voltammetry (LSV) curve for $NO_3^-$RR in acidic media (pH 1) shows a much positive onset potential than that in neutral/alkaline media, and the current density in acid is always higher than that for neutral and alkaline media at the same potential. However, with potential being more negative, the current density for alkaline medium surpasses that for neutral medium due to the slow reaction kinetics in neutral medium, as evidenced by the more sluggish kinetics shown in Tafel slopes (Fig. 1c) and Nyquist plots (Supplementary Fig. 1 and Table 1). In addition, the Nyquist plot for acidic medium displays a lower ohmic resistance loss because of higher ionic conductivity than neutral medium[29]. Moreover, the activation energies ($E_a$) of $TiO_2$ for $NO_3^-$RR were calculated by studying the temperature-dependent reaction kinetics. Supplementary Fig. 2a−c shows the LSV curves in 0.5 M $NO_3^-$ with different pHs in the temperature range of 293 K to 303 K and $E_a$ values can be obtained by referring to the previous report[30]. When the potential is −0.25 V, the $E_a$ for pH 1 is 10.7 kJ $mol^{-1}$, much smaller than that for pH 7 and pH 13 (Supplementary Fig. 2d), suggesting a lower energy barrier for $NO_3^-$RR in acidic media.

All the liquid products of $NO_3^-$RR such as $NH_3$, $N_2H_4$ and $NO_2^-$ are detected by the UV−Vis spectrophotometer and the corresponding calibration curves are provided in Supplementary Figs. 3−7. We calculated the $NH_3$ FE at different potentials in three media after electrolysis (Supplementary Fig. 8). At pH 1, $TiO_2$ shows high $NH_3$ FE (from

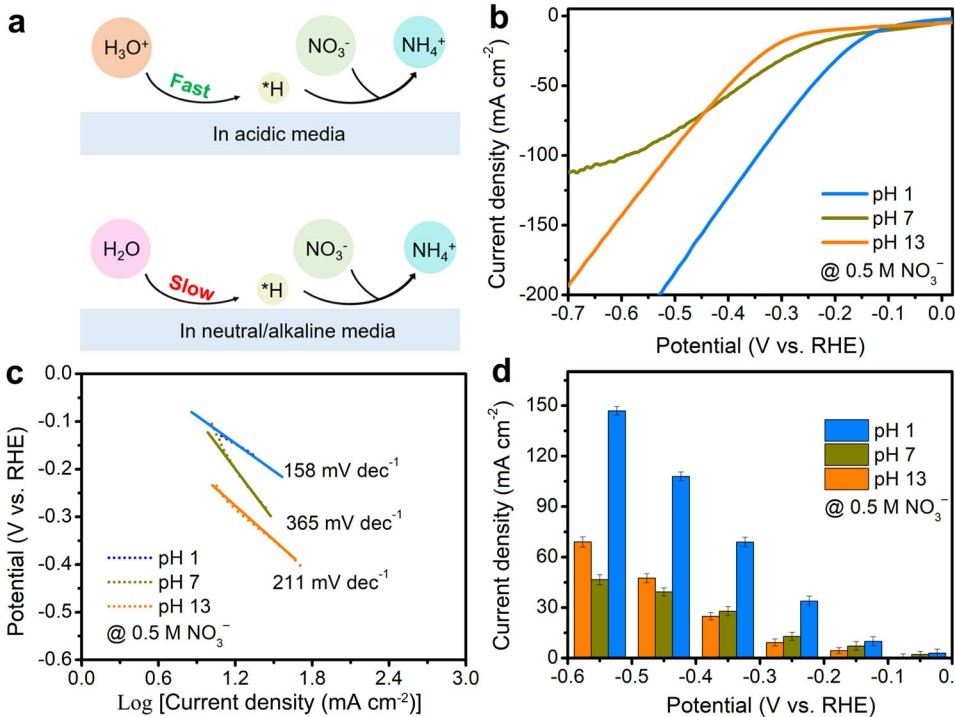

**Fig. 1 | Acidic media enables higher kinetics for $NO_3^-$RR of $TiO_2$. a** Pathways for *H production in $NO_3^-$RR process under different conditions. The acidic conditions offer abundant protons for continuous hydrogenation reactions of $NO_3^-$. **b** LSV curves and (**c**) Tafel curves of $TiO_2$ nanosheet for $NO_3^-$RR with 0.5 $NO_3^-$ in the electrolyte at different pH values. **d** Partial current density for $NH_3$ production of $TiO_2$ nanosheet at different pH values and potentials. Error bars are determined from three replicate trials at different potentials.

66% to 78.5%) at potentials in the range of −0.05 V and −0.55 V versus reversible hydrogen electrode (vs. RHE), as shown in Supplementary Fig. 9. However, NH₃ FE in neutral/alkaline media are very low (around 20%) at the first potential, increasing with the potential being more negative until −0.45 V. The highest NH₃ FE in acidic conditions is 78.5% at −0.25 V, while it is 74.3% at −0.45 V for neutral medium and 68.8% at −0.45 V for alkaline medium. The NH₃ partial current densities at different potentials and pH values of TiO₂ are shown in Fig. 1d. It is clear that TiO₂ consistently exhibits a higher NH₃ yield rate at pH 1 than that at pH 7 and pH 13 at the same applied potential, suggesting that acidic media enables a higher NH₃ yield.

In order to investigate the pH-dependent influences on the reaction pathways, we also examined the evolution of free energy plots at pH values of 1, 7, and 13. Our analysis of the pH-dependent influences on free energies of reaction intermediates is elucidated through the role of H⁺. It becomes evident that NO₃⁻RR exhibits favorable energetics when H⁺ ions are readily available within an acidic medium. This trend is substantiated by the energy evolution diagram for NO₃⁻RR (Supplementary Fig. 10). As the pH increases to 7 and 13, a concomitant increase in the free energies for NO₃⁻RR is observed due to the sluggish kinetics of H⁺ produced from additional water dissociation. The foregoing analysis further suggests that an acidic environment may be more conducive to NO₃⁻RR for TiO₂. All these results demonstrate that the acidic medium enables faster hydrogenation kinetics for NO₃⁻ RR

and more energy-efficient NH₃ synthesis for TiO₂ compared to neutral and alkaline media.

Apparently, the NH₃ FE of 78.5 % achieved with the TiO₂ catalyst in the acidic solution is still far from satisfactory and needs further improvement for practical use. Metallophthalocyanine dye is a common molecular catalyst for CO₂ reduction with poor HER performance and can operate stably under acidic conditions[31–33]. Fe plays an essential role in biological nitrite reductases to produce NH₃ through the photosynthetic nitrate assimilation pathway[34,35]. Fe phthalocyanine (FePc) is thus a promising NO₃⁻RR catalyst. Through the simple wet chemical process, we prepared FePc/TiO₂ hybrid where Pc molecules are attached to TiO₂ surface via chemical bond and van der Waals forces. Scanning electron microscope (SEM) images of TiO₂ and FePc/TiO₂ show the nanosheet structure (Supplementary Fig. 11). The mappings further reveal the uniform distribution of Ti, O, Fe, N and C within FePc/TiO₂ (Supplementary Fig. 12) and the energy dispersive X-ray spectroscopy (EDS) spectra suggest the co-existence of these elements with a weight percent of 0.73% for Fe element in FePc/TiO₂ (Supplementary Fig. 13). The X-ray diffraction (XRD) of FePc/TiO₂ shows similar peaks as TiO₂ due to the low content of FePc (Fig. 2a). Raman spectra of FePc/TiO₂ (Fig. 2b) shows typical signals for both FePc and TiO₂[36], which is further verified by the FTIR measurement (Fig. 2c)[37]. A series of FePc/TiO₂ with different treatment time of 3, 6, and 9 h (donated as FePc/TiO₂-x, x = 1, 2, and 3, respectively) are

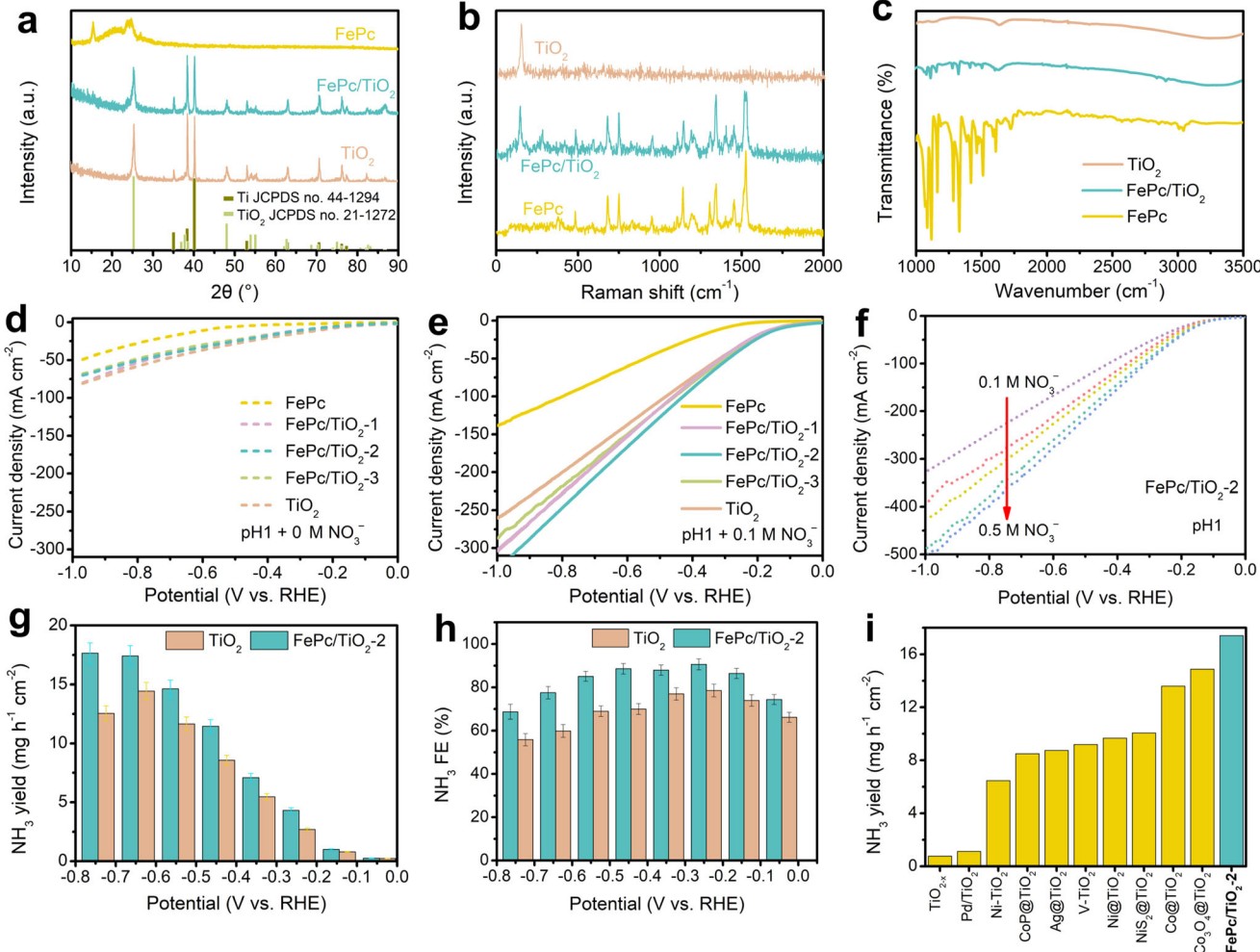

**Fig. 2 | Characterization and electrocatalytic NO₃⁻RR performance of TiO₂ and FePc/TiO₂.** **a** XRD patterns, (**b**) Raman spectra, and (**c**) FTIR spectra of TiO₂, FePc/TiO₂ and FePc. The LSV curves of TiO₂, FePc/TiO₂-x (x = 1, 2 and 3) and TiO₂ in (**d**) 0.1 M HClO₄ and (**e**) 0.1 M HNO₃, and (**f**) electrolytes (pH 1) with different NO₃⁻ concentrations. **g** The NH₃ yield and (**h**) NH₃ FE of TiO₂ and FePc/TiO₂-2 in 0.5 M NO₃⁻ at different potentials. Error bars are determined from three replicate trials at different potentials. **i** Comparison of potential and NH₃ FE for NO₃⁻RR between FePc/TiO₂−2 and the reported electrocatalysts.

prepared for comparison and their characterizations are provided in Supplementary Fig. 14. All of the samples including FePc, $TiO_2$ and FePc/$TiO_2$-x show poor catalytic HER activity with large overpotentials in acidic media (pH 1) without $NO_3^-$, as verified by the LSV curves in Fig. 2d, while the catalytic current density increases significantly with the presence of $NO_3^-$ (Fig. 2e). Notably, FePc/$TiO_2$-x shows an increased current density compared to $TiO_2$ and FePc/$TiO_2$-2 displays the largest current density among the five samples. In contrast, FePc shows the lowest current density at fixed potentials. With the increased concentration of $NO_3^-$, the current density of FePc/$TiO_2$-2 can be further improved (Fig. 2f). The $NH_3$ yield and corresponding FE at $NO_3^-$ different concentrations of FePc/$TiO_2$-2 were further obtained based on the established calibration curves for different $NO_3^-$ electrolytes (Supplementary Fig. 15). The $NH_3$ formation rate of FePc/$TiO_2$-2 shows an increased trend with the increased $NO_3^-$ concentrations from 0.1 M to 0.5 M at all potentials (Supplementary Fig. 16a). When the $NO_3^-$ concentration is up to 2 M, the $NH_3$ formation rate still significantly increases and reaches 22.5 mg h$^{-1}$ cm$^{-2}$ at −0.75 V. Besides, the $NH_3$ FE also shows similar increased trends with increased $NO_3^-$ concentration at each potential (Supplementary Fig. 16b), suggesting that the high $NO_3^-$ concentration fascinates the $NH_3$ formation. Specifically, the maximum $NH_3$ FE values for 0.1 M, 0.2 M, 0.3 M, 0.4 M, and 0.5 M $NO_3^-$ electrolytes using FePc/$TiO_2$-2 are determined as 77.0%, 79.2%, 83.2%, 87.9%, and 90.6%, respectively. The $NH_3$ FE in 2 M $NO_3^-$ is 92.7%, close to that in 0.5 M $NO_3^-$ electrolytes. Therefore, we chose 0.5 M for $NO_3^-$RR in the next explorations. After electrolysis at different potentials (Supplementary Fig. 17), we determined the concentration of production in the diluted electrolyte to calculate their FEs. The FePc/$TiO_2$-2 shows a larger $NH_3$ yield rate and FE than those for $TiO_2$ at each potential (Fig. 2g, h). Additionally, the $NH_3$ yield (partial current density) of FePc/$TiO_2$-2 steadily increases with the potential being more negative and reaches 17.4 mg h$^{-1}$ cm$^{-2}$ (219.7 mA cm$^{-2}$) at −0.65 V (Supplementary Fig. 18). The FePc/$TiO_2$-2 shows high $NH_3$ FE above 85% at −0.15 V to −0.55 V. It reaches the maximum value (90.6%) at −0.25 V and keeps at 68.7% even at −0.75 V. The $NO_2^-$ byproduct was detected at the beginning on both FePc/$TiO_2$-2 and $TiO_2$ catalysts electrodes, but the $NO_2^-$ FEs showed a similar decreased trend with more negative potentials (Supplementary Fig. 19). Besides, the UV–Vis. adsorption curves of the electrolytes collected at different potentials are totally overlapped with that of blank standard solution (Supplementary Fig. 20), indicating that almost no hydrazine was formed during $NO_3^-$RR for both FePc/$TiO_2$-2 and $TiO_2$. Double layer capacitance ($C_{dl}$) of FePc/$TiO_2$-2 is slightly larger than that of $TiO_2$, indicative larger electrochemical active surface area on FePc/$TiO_2$-2 for efficient catalysis (Supplementary Fig. 21)[38]. The FePc is active for $NO_3^-$RR but only delivers the maximum $NH_3$ FE of 76.0% at −0.45 V (Supplementary Fig. 22). The mass activity of FePc/$TiO_2$ with different FePc loading masses were also studied. FePc/$TiO_2$-2 shows a higher geometrical area and mass normalized $NH_3$ yield than FePc/$TiO_2$-1 and FePc/$TiO_2$-3 (Supplementary Fig. 23a, b), revealing that higher or lower FePc mass loading would lead to the decreased $NH_3$ yield. Besides, it shows similar peak FE values for $NH_3$ formation with 88.7%, 90.6%, and 87.4% for FePc/$TiO_2$-1, FePc/$TiO_2$-2, and FePc/$TiO_2$-3 (Supplementary Fig. 23c), respectively, suggesting that the catalyst loading shows limited impact on the $NH_3$ FE of $NO_3^-$RR. All these results demonstrate the enhanced electrochemical $NO_3^-$RR performance of FePc/$TiO_2$-2. As far as we know, the FePc/$TiO_2$-2 exhibits higher $NH_3$ yield than most reported $TiO_2$-based catalysts, as shown in Fig. 2i and Supplementary Table 2. Though there is still a gap in performance between FePc/$TiO_2$-2 and the state-of-the-art $NO_3^-$RR electrocatalysts performed in alkaline media with high $NO_3^-$ concentrations up to 1 M, it can be confirmed that the precious-metal-free FePc/$TiO_2$-2 shows higher $NH_3$ synthesis performance than RuCu[39] and $Fe_2M$-trinuclear-cluster metal−organic frameworks[40] catalysts recently reported in acid.

To determine the nitrogen source in the conversion of $NO_3^-$-to-$NH_3$ and the amount of produced $NH_3$, we employed $^{15}NO_3^-$ and $^{14}NO_3^-$ as the feedstock and detected the $NH_3$ by $^1H$ nuclear magnetic resonance (NMR) method. We first built the calibration curve using the standard $(^{14}NH_4)_2SO_4$ and $(^{15}NH_4)_2SO_4$, as displayed in (Supplementary Fig. 24). Then, we conducted the electrolysis at −0.45 V for 0.5 M $^{15}NO_3^-$ and $^{14}NO_3^-$ electrolytes, respectively. The electrolytes were collected for further analysis. Typical double and triple peaks appear in NMR spectra for $^{15}NH_4^+$ and $^{14}NH_4^+$, respectively (Fig. 3a). Additionally, the $NH_3$ yield rate and FE at −0.45 V obtained from the NMR method are finally recorded at 11.2 mg h$^{-1}$ cm$^{-2}$ and 85.0% with $^{14}NO_3^-$ and at 11.8 mg h$^{-1}$ cm$^{-2}$ and 88.7% with $^{15}NO_3^-$, which are close to the UV−Vis. results (Fig. 3b), indicative of the reliability of the experimental data for $NH_3$ determination. Meanwhile, the maximum $NH_3$ FE at −0.25 V maintains a stable value with prolonged reaction time (Fig. 3c). To investigate the long-term stability, we extended the electrolysis of FePc/$TiO_2$-2 to a duration of ~24 hours in 60-mL electrolyte of 0.5 M $NO_3^-$ (pH 1). The current density exhibits a slight decrease during the initial two hours, followed by a sustained and nearly constant trend in the subsequent time period (Supplementary Fig. 25). The $NH_3$ FE recorded at different times remains almost stable and only decreases slightly in the final 6 hours from 86.6% to 81.9%. These results indicate the good electrochemical stability of FePc/$TiO_2$. Furthermore, the post-test FePc/$TiO_2$-2 is thoroughly characterized. SEM image confirms the intact nanosheet structure (Supplementary Fig. 26). XRD patterns of FePc/$TiO_2$−2 after electrolysis at different potentials show nearly the same diffraction peaks as the original one (Fig. 3d), indicating the good structure stability of FePc/$TiO_2$−2. EDS mappings and FTIR analyses reveal the presence of FePc in the tested electrode (Supplementary Fig. 27). We also collected the XPS spectra of FePc/$TiO_2$−2 after electrolysis for 1 h at different potentials for comparison. As shown in Fig. 3e, with the potential being more negative, the typical peaks for $Ti^{4+}$ in Ti 2$p$ region slightly shift toward smaller binding energies, indicating more $Ti^{x+}$ with a low valence state may generate during the $NO_3^-$RR[41]. In the N 1$s$ spectra, a new peak around 408.5 eV assigned to absorbed $NO_3^-$ and the peak around 397 eV assigned to the surface $NH_3$ species are observed during the electrolysis (Fig. 3f)[42,43]. However, with the potential being more negative, the peak assigned to Fe-N in FePc becomes weaker[44]. To investigate whether the catalyst dissolved in the solution, we conducted inductively coupled plasma-mass spectrometry (ICP-MS) measurement to the solution after electrolysis. Only 5.17 wt% of Fe dissolved in the first 4 hours, and this figure increases to about 20 wt% after 24-hour electrolysis (Supplementary Fig. 28). It should be noted that Ti element is also detected in the solution, suggesting the inevitably partial dissolution of the overall FePc/$TiO_2$−2 electrode in the acidic conditions after long-term electrolysis, which may be associated with the slight loss of electrocatalytic $NO_3^-$RR activity.

To identify the $NO_3^-$ absorbed on Fe site, FePc/$TiO_2$−2 was immersed in the electrolyte solution for 24 hours, and then collected the XPS spectra before and after immersion for further analysis. The typical peak for $NO_3^-$ appears in the spectra of FePc/$TiO_2$−2 after immersion, and the peak for Fe 2$p_{3/2}$ region shifts toward higher binding energy compared to that before the immersion (Supplementary Fig. 29), indicating the possible interaction between $NO_3$ and Fe centers after adsorption. To further elucidate the role of the active site of FePc/$TiO_2$ in $NO_3^-$RR, we first conducted SCN$^-$ intoxication experiments to block Fe species of FePc/$TiO_2$−2 because of the strong affinity of SCN$^-$ with Fe species[45,46]. It was found that the $NO_3^-$RR performance of FePc/$TiO_2$-2 decreased significantly in terms of current density, $NH_3$ FE and $NH_3$ yield after adding KSCN (Supplementary Fig. 30), indicating the active Fe center for $NO_3^-$RR. In-situ FTIR was also employed to detect the reaction intermediates adsorbed on the catalysts' surface during the $NO_3^-$RR electrolysis. When potential decreases from 0 to −0.55 V, several peaks appear and increase in intensity for FePc/$TiO_2$-2

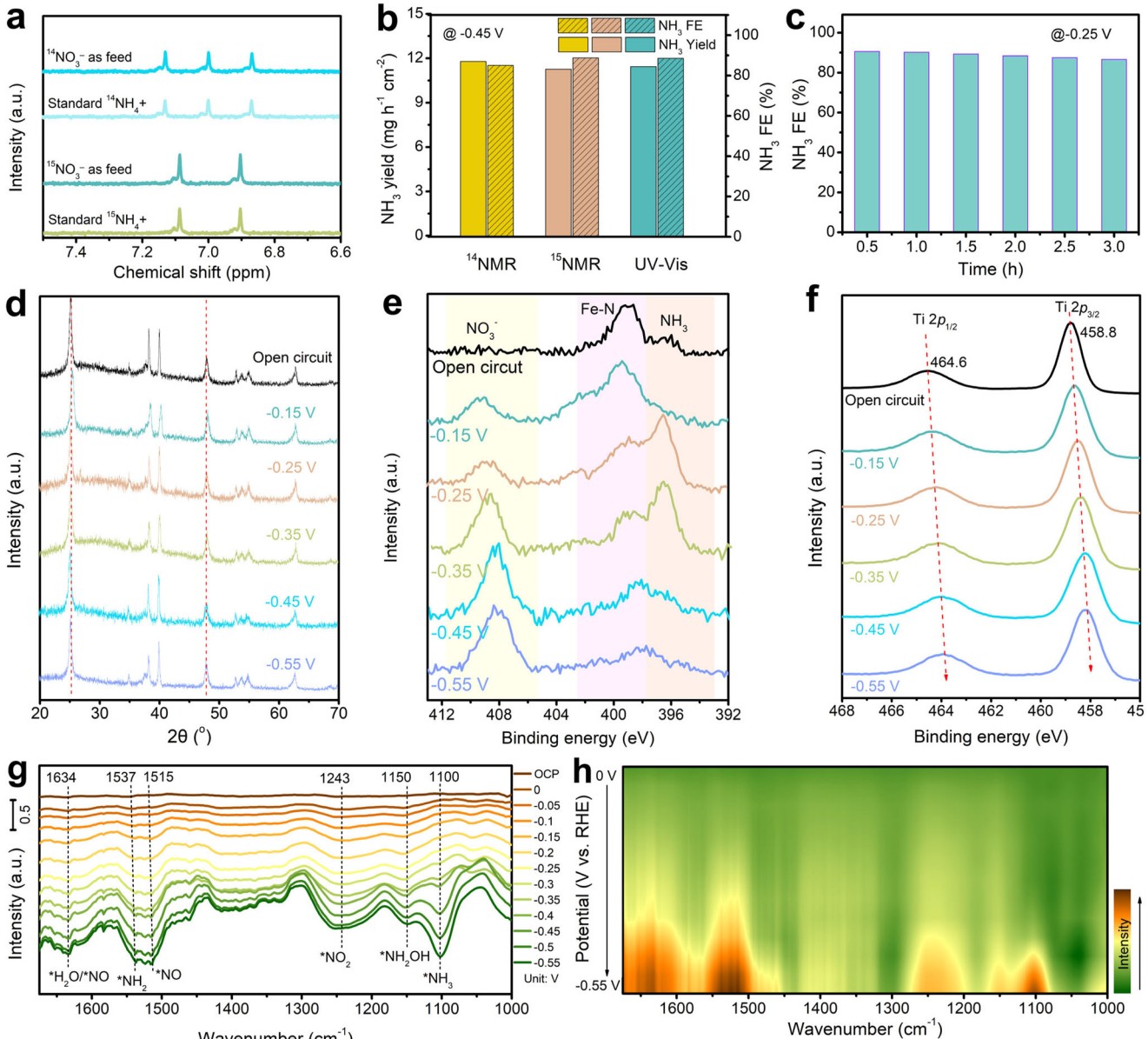

**Fig. 3 | Analysis of the FePc/TiO₂ during NO₃⁻RR electrolysis. a** ¹H NMR spectra of the electrolytes after the measurements and the referenced spectrum of (¹⁵NH₄)₂SO₄. **b** The NH₃ yield rate and FE of FePc/TiO₂-2 at −0.45 V measured by NMR and UV–Vis methods for a quantitative comparison. **c** The NH₃ FE of FePc/TiO₂-2 after electrolysis at various time at −0.25 V in acidic conditions. **d** The XRD patterns, (**e**) XPS spectra in Ti 2p region and (**f**) XPS spectra in N 1s region of FePc/TiO₂-2 before and after electrolysis at different potentials. **g** In-situ FTIR spectra of FePc/TiO₂-2 working at different potential for NO₃⁻RR. **h** The 2D FTIR contour map of FePc/TiO₂-2 for NO₃⁻RR.

(Fig. 3g, h). The characteristic peak at 1634 cm⁻¹ is attributed to the bending vibration of the adsorbed water involved in the NO₃⁻RR in the solution of the thin layer or the NO intermediate due to the overlap of peaks[47]. The band at 1537 cm⁻¹ could be assigned to NH₂ bending, while the band at 1515 cm⁻¹ is ascribed to the vibration peak of NO[48,49]. Meanwhile, the absorption bands centered at 1243 cm⁻¹ are assigned to symmetric and antisymmetric stretching vibration of the NO₂ group. The bands around 1150 cm⁻¹ and 1100 cm⁻¹ are attributed to the H-N stretching vibration of hydroxylamine (NH₂OH) and produced NH₃[50], respectively. Similar peaks were also observed on FePc (Supplementary Fig. 31). These observation reveals the possible NO₃⁻RR pathway as follows: NO₃⁻ → *NO₃ → *NO₂ → *NO → *NOH → *NH₂OH → *NH₃.

Density functional theory calculations were carried out to gain insight into the NO₃⁻ reduction reaction on TiO₂ and FePc/TiO₂ systems. It was discovered that the FePc molecule prefers to lie on the surface of TiO₂ with its basal plane parallel to the TiO₂(101) surface. A

clear chemical bond (~2.28 Å in length) is formed between the Fe site and the lattice oxygen on the TiO₂ surface within the FePc/TiO₂ composite (Supplementary Fig. 32). Conversely, the remaining portion of the FePc primarily exhibits van der Waals interactions with the TiO₂ substrate. Further analysis, involving charge density difference assessments and density of states examinations, substantiates a considerable charge transfer (0.12 e⁻) and robust Fe-O 3d-2p orbital overlap below the Fermi level between FePc and TiO₂ (Supplementary Fig. 33), thus confirming a genuine chemical interaction rather than merely van der Waals interactions. To assess the stability of the Fe−O chemical bond during NO₃⁻ reduction, we performed relative energy calculations, which can quantifiably measure the interactions between FePc and TiO₂. Our findings reveal an interaction energy of −2.55 eV between FePc and TiO₂, with this value remaining consistently close to −2.5 eV even after the adsorption of intermediate species on the Fe site (Supplementary Fig. 34). This consistency demonstrates the enduring

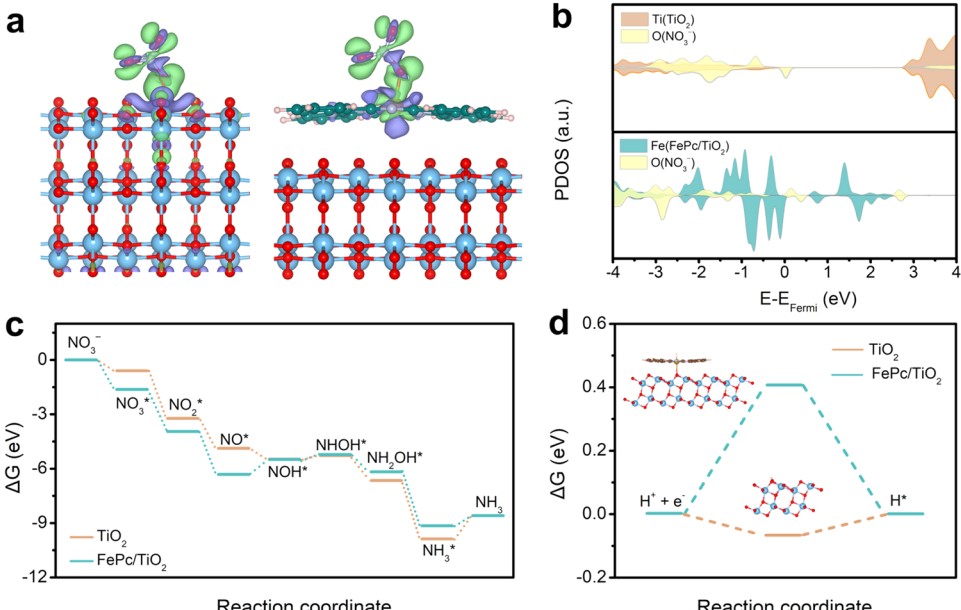

**Fig. 4 | Theoretical calculation for NO₃⁻RR reaction pathway over TiO₂ and FePc/TiO₂. a** Charge density difference for NO₃⁻ adsorption on the TiO₂ (left) and FePc/TiO₂ (right). Blue, red, pink and green balls represent Ti, O, H, and Fe atoms, respectively. **b** The PDOS for NO₃⁻ adsorbed on the TiO₂ and FePc/TiO₂. Free energy diagram for (**c**) NO₃⁻ reduction and (**d**) HER on the TiO₂ and FePc/TiO₂.

and robust nature of the chemical interactions with the FePc/TiO₂ composite. NO₃⁻ is preferably adsorbed on the Fe site with a larger adsorption energy of 1.48 eV compared to Ti (Supplementary Fig. 35), implying that Fe is more likely to be the active site for NO₃⁻RR. Charge density difference analysis reveals significant charge transfer between NO₃⁻ and the Fe site, as depicted in Fig. 4a. This is further corroborated by the density of states analysis, which demonstrates strong *d-p* orbital overlap at a more negative energy level below the Fermi level for Fe-NO₃ compared to Ti-NO₃ (Fig. 4b), indicating a more robust covalent bond component in the Fe-NO₃. Figure 4c shows the thermodynamics of the NO₃⁻ reduction reaction on TiO₂ and FePc/TiO₂ surfaces. Supplementary Figs. 36 and 37 display the adsorption configuration of different N-containing intermediates. From the Gibbs free energy profile, it is clear that the rate-limiting step for NO₃⁻ reduction on the FePc/TiO₂ surface is the formation of the NOH* intermediate, which requires an energy input of 0.74 eV. FePc also exhibits the reduction of *NO to *NOH is the rate-determining step with an energy barrier of 0.81 eV (Supplementary Fig. 38a, b). It means that NO is an important reaction intermediate, which agrees with our in-situ FTIR spectra observations. In contrast, for the pristine TiO₂ system, the reaction-determining step is NH₃ desorption, which requires at least 1.29 eV of energy input. Overall, the FePc/TiO₂ composite exhibits higher catalytic activity for reducing NO₃⁻ than pristine TiO₂ and FePc. Remarkably, the FePc/TiO₂ catalyst shows inert activity towards HER, a side reaction with a reaction-free energy of 0.42 eV (Fig. 4d), demonstrating the superior HER resistance. Conversely, the reaction free energy of HER on pristine TiO₂ and FePc is -0.06 and 0.38 eV (Supplementary Fig. 38c), respectively.

The development of NOₓ⁻-based electrochemical battery systems holds great promise in sustainable development since it can produce high value-added NH₃ and meanwhile generate electricity[51–55]. Thanks to the high theoretical capacity and low redox potential of Zn metal[56–58], aqueous Zn–NO₃⁻ batteries have emerged as such an ideal system. Zhi's group first developed galvanic Zn–NO₃⁻ cell with an OCV of 0.81 V and a power density of 0.87 mW cm⁻², based on a Pd/TiO₂ supported on carbon cloth as the cathode[19]. Subsequently, they demonstrated that such Zn–NO₃⁻ battery with OCV of 1.22 V can be rechargeable but irreversible and cathodic water oxidation occurs

during the charging process[9]. Interestingly, Jiang et al. recently proposed a rechargeable and reversible Zn-nitrogen flow batteries and NH₃ can be oxidized to NO₂⁻ and further NO₃⁻ during charging process. However, it lost the unique advantage of turning waste into treasure and such battery only exhibits an OCV of 1.39 V and a power density of 10.0 mW cm⁻²[52]. Up to now, the highest power density of Zn–NO₃⁻ Zhou's group reports battery[59]. Benefiting from the high NO₃⁻RR activity, the galvanic Zn–NO₃⁻ battery with OCV of 0.94 V exhibits a power density as high as 70.7 mW cm⁻² with CuNi nanoparticles supported on Cu foil as the cathode and 3.5 M NaOH/0.71 M NO₃⁻ catholyte. However, all reported Zn–NO₃⁻ batteries deliver limited voltages and the power density and NH₃ yield/selectivity are still desired for this electrochemical cell system, which severely depends on the conditions of the cathodic part. For a Zn–NO₃⁻ battery, a more positive potential and a higher catalytic current for acidic NO₃⁻RR at the cathode generally contribute to a higher output power density. Compared to the reported Zn–NO₃⁻ batteries equipped with NO₃⁻RR in neutral/alkaline conditions, an alkaline-acidic hybrid Zn–NO₃⁻ battery (AAHZNB) is highly attractive to offer a large power density with high NH₃ yield.

Inspired by the above considerations, we assembled such an AAHZNB battery with FePc/TiO₂-2 as the cathode (in the acidic electrolyte) and Zn plate (in the alkaline electrolyte) as an anode. A bipolar membrane is used to separate the cathodic and anodic chambers (Fig. 5a). The Zn–NO₃⁻ battery delivers a stable OCV of 1.99 V (Fig. 5b), higher all reported values of Zn–NO₃⁻ batteries up to now (Supplementary Table 3). The high OCV is ascribed to the intrinsic potential difference of Zn oxidation and NO₃⁻RR as well as the pH difference between the two chambers separated by a bipolar membrane, which can efficiently avoid the direct neutralization of acid and alkali. Supplementary Fig. 39 shows LSV curves of anodic oxidation reaction of Zn plate and the cathodic NO₃⁻RR using FePc/TiO₂ in different media at potentials versus Ag/AgCl reference electrode. It is clear that the acidic NO₃⁻RR shows a larger potential gap than that for neutral and alkaline media. NH₃ product formed in the cathodic compartment was identified after discharging at different voltages (Supplementary Fig. 40). AAHZNB during the discharge process achieves a peak FE of 88.2% at 0.7 V and a high NH₃ yield of 12.3 mg h⁻¹ cm⁻² at 0.4 V (Fig. 5c). Importantly, AAHZNB exhibits an exceptional peak power density of

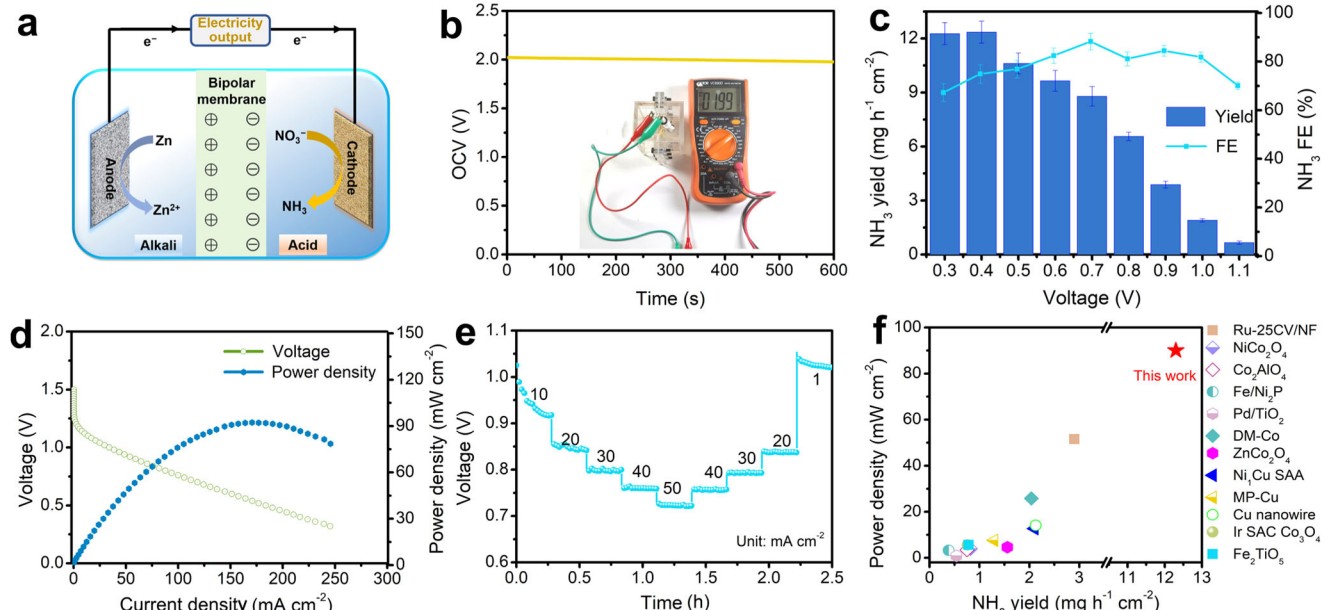

**Fig. 5 | High voltage Zn–NO$_3^-$ batteries for simultaneous NO$_3^-$ conversion to NH$_3$ and electricity supply. a** The schematic diagram of the AAHZNB with FePc/TiO$_2$-2 as cathode and Zn foil as anode. **b** The OCV of the AAHZNB. **c** The NH$_3$ yield and FE after discharge at different voltages of the AAHZNB. Error bars are determined from three replicate trials at different voltages. **d** The discharge curve of AAHZNB and the corresponding power density. **e** The multi-step chronoamperometric curve of the AAHZNB for discharging at different current densities. **f** Comparison of power density between the AAHZNB and the reported Zn–NO$_3^-$ batteries (CV is cyclic voltammetry, NF is nickel foam, DM is dense membranous, SAA is single atom alloy, MP is metastable phase and SAC is single atom catalyst).

91.4 mW cm$^{-2}$ (Fig. 5d). We also collected the LSV curve flow Zn–NO$_3^-$ battery. It is found that the flow Zn–NO$_3^-$ battery shows a peak power density of 93.55 mW cm$^{-2}$, which is very close to that obtained from a static mode with stirring (Supplementary Fig. 41). The rate performance of the AAHZNB was also tested. Figure 5e shows the ladder-shaped discharging curves at different current densities and each step exhibits stable discharging plateaus. After discharging at a current density up to 50 mA cm$^{-2}$, the voltages are well recovered at current densities of 40, 30, and 20 mA cm$^{-2}$, indicating the high stability of AAHZNB. Supplementary Figure 42 shows the chronopotentiometry curves and the NH$_3$ FE at different times. The acid-base hybrid Zn–NO$_3^-$ battery shows slightly decreased current density at the first 2 h at 0.9 V and maintains stability in the next 8 h. Besides, the NH$_3$ FE of the cathode changes from the initial 84.7% to 81.8% after 10-h electrolysis. As such, we can conclude that AAHZNB employed by the FePc/TiO$_2$-2 could achieve electricity output with the maximum power density of 91.4 mW cm$^{-2}$ and NH$_3$ production rate of 12.3 mg h$^{-1}$ cm$^{-2}$, which are highest among the Zn–NO$_x^-$ batteries reported so far (Fig. 5f). Besides, it even surpasses some of the reported Zn-H$_2$O batteries, solid-state Zn-air batteries (ZAB), direct urea fuel cells (DUFCs), direct hydrazine fuel cells (DHzFCs), direct formic acid fuel cells (DUFCs), as summarized in Supplementary Table 4.

The high performance of FePc/TiO$_2$-2 for acidic NO$_3^-$RR and AAHZNB provides more opportunities for the practical applications. One of the possible applications is the environmental sulfur recovery driven by high power density of AAHZNB. H$_2$S/S$^{2-}$ are common pollutants in the exhaust gas and sewage of industry. Electrochemical oxidation of S$^{2-}$ or the H$_2$S splitting as valuable sulfur species can be combined with cathodic hydrogen evolution, which holds great significance environmentally and economically[60]. One step further, the hydrazine (N$_2$H$_4$) can be employed to replace Zn anodes in the nitrate-based battery due to its low oxidation potential[61], realizing a fuel cell treating both pollutes and producing electricity simultaneously. Figure 6a shows the picture of an electrolyser composed of a cathodic hydrogen evolution and anodic sulfur oxidation reaction (SOR) with commercial Pt/C as bifunctional catalysts successfully powered by the

AAHZNB with the current density 35.6 mA cm$^{-2}$. With electrolysis going, a gradual rise of UV–Vis absorption bands at 300 nm is observed, indicating the generation of short-chain S$_2^{2-}$ in the anolyte during electrolysis (Fig. 6b)[60]. The oxidation product can be efficiently obtained through the acid treatment of the electrolyte and XRD pattern for the collected yellow powder corresponds to the elemental sulfur (Supplementary Fig. 43). Figure 6c shows the schematic diagram of the proposed N$_2$H$_4$-NO$_3^-$ fuel cell with FePc/TiO$_2$-2 as the cathode. We used commercial Pt/C as an anodic catalyst and found that the onset potential is about −1 V vs. Ag/AgCl and the current density reaches saturation in 1 M KOH + 0.3 M N$_2$H$_4$ (Fig. 6d). The onset potential for HzOR is negative than that for NO$_3^-$RR, indicating that the N$_2$H$_4$-NO$_3^-$ fuel cell is workable. The assembled N$_2$H$_4$-NO$_3^-$ fuel cell shows a discharge voltage of 0.75 V at 1 mA cm$^{-2}$ and a peak power density of 11.5 mW cm$^{-2}$ (Fig. 6e). The NH$_3$ FE keeps at around 80% when discharged at different voltages (Fig. 6f and Supplementary Fig. 44). All these studies unveil the practical applicability of NO$_3^-$-based fuel cells with great energy and environmental significance and further introduce NO$_3^-$ as a potential member in fuel cell systems.

## Discussion

Thanks to the advantages of abundant protons supply and fast kinetics of NO$_3^-$RR in acidic conditions, we have successfully reported the high-efficiency NH$_3$ electrosynthesis using the FePc/TiO$_2$ nanosheet. Such catalyst delivers an NH$_3$ yield rate of 17.4 mg h$^{-1}$ cm$^{-2}$ and a FE of 90.6% for acidic NO$_3^-$-to-NH$_3$ conversion at pH 1. The introduction of FePc strengthens the adsorption of the N-containing intermediate and reduces the energy barrier of the rate-determining step, thus promoting the NO$_3^-$RR performance of TiO$_2$. Furthermore, the alkaline-acidic hybrid Zn–NO$_3^-$ battery was developed for simultaneous NH$_3$ and electricity generation with a high open-circuit voltage of 1.99 V and an impressive power density of 91.4 mW cm$^{-2}$, which can be applied in the environmental sulfur recovery. Moreover, the developed all-pollute N$_2$H$_4$-NO$_3^-$ fuel cell can simultaneously eliminate N$_2$H$_4$/NO$_3^-$and supply electricity. This work not only demonstrates the attractive potential of acidic

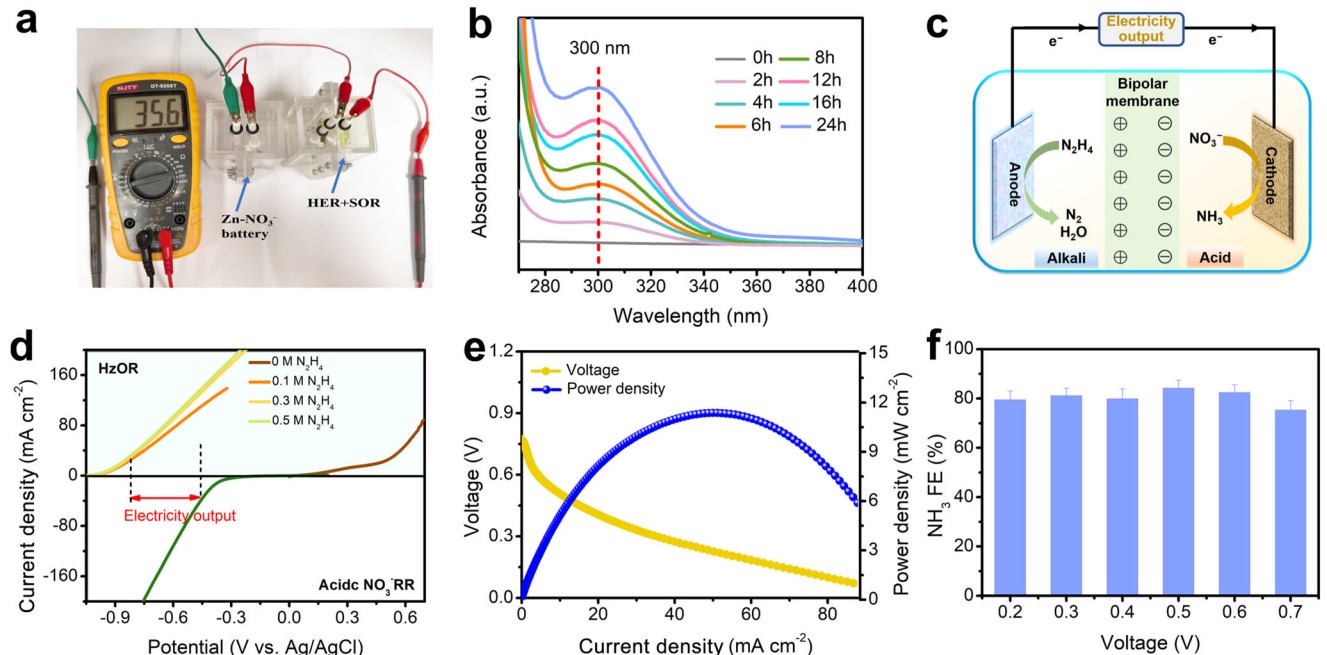

**Fig. 6 | AAHZNB for environmental sulfur recovery and $N_2H_4$-$NO_3^-$ fuel cell for electricity generation and $NH_3$ synthesis. a** The picture of the AAHZNB to power anodic SOR and cathodic HER. **b** The UV−Vis. adsorption curves for diluted electrolyte after the sulfur oxidation reaction driven by the AAHZNB for different time. **c** The schematic diagram of proposed $N_2H_4$-$NO_3^-$ fuel cell with FePc/TiO$_2$-2 as the cathode. **d** The LSV curves for the HzOR of Pt/C and the acidic $NO_3^-$RR of FePc/TiO$_2$-2. **e** The discharge curve of $N_2H_4$-$NO_3^-$ fuel cell and the corresponding power density. **f** The $NH_3$ FE at the cathode of the $N_2H_4$-$NO_3^-$ fuel cell when discharging at different voltages. Error bars are determined from three replicate trials at different voltages.

$NO_3^-$RR for $NH_3$ electrosynthesis but broadens the $NO_x^-$-based electrochemistry in Zn battery and fuel cells.

## Methods

### Materials

Ammonium sulfate (($NH_4$)$_2SO_4$, 99.95%), hydrazine monohydrate ($N_2H_4$·$H_2O$, >98%), deuterium oxide ($D_2O$, 99.9 atom % D), ethanol ($C_2H_5OH$, 99.5%) nitric acid ($HNO_3$, 70%), Perchloric acid ($HClO_4$, 70%) sulfuric acid ($H_2SO_4$, >95%), sodium hydroxide (NaOH, >98%), potassium hydroxide (KOH, >95%), potassium nitrate ($KNO_3$, 99.0%), potassium nitrate−$^{15}$N ($K^{15}NO_3$, 99 atom%), sodium hypochlorite solution (NaClO, available chlorine 4.0 %), ammonia sulfate-$^{15}$N ($^{15}NH_4$)$_2SO_4$, 99 atom%), potassium nitrite ($KNO_2$, 99%), iron phthalocyanine (FePc, 97%), hydrogen peroxide ($H_2O_2$, 30%), p-dimethylaminobenzaldehyde ($C_9H_{11}NO$), salicylic acid ($C_7H_6O_3$, 99%), ethanol ($C_2H_5OH$, 99.7%), trisodium citrate ($C_6H_5Na_3O_7$, 98%), commercial Pt//C (Pt 10%), para(dimethylamino) benzaldehyde (($CH_3$)$_2NC_6H_4CHO$, 99%), (1-Naphthyl) ethylenediamine dihydrochloride ($C_{12}H_{14}N_2$·2HCl, 98%), sulfanilamide ($C_6H_8N_2O_2S$, >99%), N,N-Dimethylformamide (DMF, 99.5%) and sodium nitroferricyanide dehydrate ($C_5FeN_6Na_2O$·$2H_2O$, 99%) were purchased from Aladdin (Shanghai) Chemistry Co., Ltd. Fumasep FBM-PK membranes were purchased from Fuel Cell Store. Flow battery equipment was purchased from Shanghai Chuxi Co., Ltd.

### Synthesis of $TiO_2$ and FePc/$TiO_2$

Typically, Ti mesh were washed by acetone, ethanol, and diluted hydrochloric acid to remove surface impurities. Then, Ti mesh (2 cm × 3 cm) was put into 5 M NaOH aqueous solution (40 mL) in a 50-mL Telfon autoclave and then heated at 180 °C for 10 h. After the autoclave was cooled down naturally, the Ti mesh was taken out, washed with deionized water several times, and dried at 60 °C in a dry cabinet. Such Ti mesh was then immersed in 1.2 M HCl for 4 h to ensure that $Na^+$ were completely replaced by $H^+$. Then, this sample was annealed at 450 °C for 1 h to obtain $TiO_2$ with loading mass of

close to 0.5 mg cm$^{-2}$. In order to prepare FePc/$TiO_2$-x, firstly, FePc (5 mg) was dissolved into a 10 mL DMF. Then, the $TiO_2$ was immersed into 50 mL of the above-mixed solution and kept at 90 °C for 3 h under continuous agitation and finally washed by deionized water to obtain the catalysts (denoted as FePc/$TiO_2$-1). The FePc/$TiO_2$-2 and FePc/$TiO_2$−3 were prepared by adjusting reaction time to 6 h and 9 h, respectively. The loading mass of FePc/$TiO_2$−1, FePc/$TiO_2$−2 and FePc/$TiO_2$−3 are estimated as 0.512 mg cm$^{-2}$, 0.526 mg cm$^{-2}$, 0.531 mg cm$^{-2}$, respectively.

### Characterization

The crystalline, morphologies and microstructures of samples were investigated by XRD using a Bruker D2 Phaser diffractometer with Cu Kα irradiation ($\lambda = 1.54$ Å) and field-emission scanning electron microscopy (FEI Quanta 450 FEG). The surficial chemical states and compositions of the as-obtained products were investigated by XPS (ESCALB 250) with an Al Kα X-ray beam ($E = 1486.6$ eV). The C 1s peak with the binding energy of 284.8 eV was used as the calibration standard. $^1$H-NMR measurements were performed on Bruker 400 MHz ASCEND AVANCE III HD Nuclear Magnetic Resonance System (NMR-400). UV−Vis spectroscopy measurements were carried out using a Dynamica Halo DB−20S UV–Vis Spectrophotometer. FTIR measurements were carried out using Spectrum Two FTIR Spectrometers by PerkinElmer. Raman spectroscopy was performed on a WITec alpha300 Raman System under excitation of 632.8 nm laser light. ICP-MS was performed on Agilent 5110.

### Electrochemical measurements

The electrochemical $NO_3^-$RR measurements were performed in a taditional H-type cell with a CHI 760E electrochemical workstation (Chenhua, China) using a three-electrode system. $TiO_2$ and FePc/$TiO_2$-x supported on Ti mesh (0.5 × 0.5 cm$^2$), Ag/AgCl (saturated KCl aqueous solution), and Pt foil (1.5 × 1.5 cm$^2$) electrode were served as working electrode (WE), reference electrode (RE), and counter electrode (CE), respectively The cathodic and anodic chambers are separated by a

Nafion 117 membrane (Dupont). The electrolyte for acidic $NO_3^-$RR is 0.1 M $HNO_3$/0.4 M $KNO_3$ (pH is about 1), while the electrolyte for HER is 0.1 M $HClO_4$ (pH 1). The electrolytes for neutral and alkaline $NO_3^-$RR are 0.1 M $K_2SO_4$/0.5 M $KNO_3$ (pH 7) and 0.1 M KOH/0.5 M $KNO_3$ (pH 13), respectively. The pH of the solution was determined when the displayed value was stable using a pH meter (PHS-3C, Leici). The electrochemical measurements were performed in the Ar atmosphere with a gas flow rate of 20 sccm and they were kept stirring (600 rpm) throughout the tests to minimize the mass transfer limitation. LSV tests were conducted with a scan rate of 5 mV s$^{-1}$. The chronoamperometry tests were performed in a typical H-type cell that contains 15-mL electrolyte for each chamber. The electrochemical impedance spectroscopy was obtained in the frequency range from 0.1 Hz to 100 kHz upon an AC voltage amplitude of 5 mV at an open-circuit potential and room conditions. All the final reference potential was converted to the reversible hydrogen electrode (RHE) using the Nernst equation: E(RHE) = E(Ag/AgCl) + 0.197 + 0.059 × pH. To prepare the Pt/C loaded electrode to catalyze SOR and HzOR, Pt/C (10 mg) and Nafion solution (50 μL 5 wt%) were dispersed in water/ethanol solvent mixture (1 mL, 1:1 v/v) by 30 min sonication to form an ink. Then catalyst ink (100 μL) was loaded on a carbon cloth (1 × 1 cm$^2$) with mass loading of 1 mg cm$^{-2}$.

## Assembly of batteries and electrochemical test

For Zn–$NO_3^-$ battery, the FePc/TiO$_2$-2 (1 cm$^2$) and Zn plate (4 cm$^2$, thickness = 0.2 mm) were used as the cathode and anode for Zn–$NO_3^-$ battery, respectively. The distance of both electrodes is 4 mm. 15-mL catholyte (0.1 M $HNO_3$ + 0.4 M $KNO_3$) and 15-mL anolyte (6 M KOH) were separated by a bipolar membrane (fumasep FBM, 4 cm$^2$). The discharge polarization curves with a scan rate of 10 mV s$^{-1}$ were collected using a CHI 760E workstation under room conditions with vigorous stirring (600 rpm). After electrolysis at different voltages for 30 min, the electrolyte solution was collected and diluted for the subsequent detection. The power density (P) of the Zn–$NO_3^-$ battery was the product of voltage and discharge current density. For the flow mode of Zn–$NO_3^-$ battery, the electrolyte in the cathode and anode was circulated by a peristaltic pump (Kamoer, F01A-STP) with a flow rate of 25 mL min$^{-1}$. The assemble of $N_2H_4$-$NO_3^-$ fuel cell follows the similar procedure except the anode is replaced by Pt/C electrode and the anodic electrolyte is 1 M KOH + 0.3 M $N_2H_4$.

## Calculation of $E_a$ for $NO_3^-$RR

To extract the $E_a$ for the $NO_3^-$RR, the LSV curves were first collected in a 0.5 M $NO_3^-$ solution at different pH values and at different temperatures. Then, $E_a$ can be determined according to the Arenius equation: j = Aexp (−$E_a$/RT), where j is the current density, A is the apparent preexponential factor, R is the ideal gas constant and T is the temperature.

## Determination of NH₃

The quantity of generated $NH_3$ was assessed utilizing the indophenol blue method[55]. All data was recorded three times to obtain the error bar. Subsequent to the $NO_3^-$RR test, a 1-mL aliquot of diluted electrolyte was extracted, neutralized, and subjected to further analysis. A solution comprising 0.625 M NaOH, 0.36 M salicylic acid, and 0.17 M sodium citrate (1.25 mL) was added to the neutralized electrolyte. Following this, 150 μL of sodium nitroferricyanide solution (10 mg mL$^{-1}$) and 75 μL of NaClO (available chlorine 4.0 wt%) solution were introduced. Following a 2-hour incubation period under ambient conditions, the UV–Vis absorption spectrum was recorded, and the absorbance value at 658 nm was determined. A concentration-absorbance calibration curve was established using a series of known standard concentrations of $(NH_4)_2SO_4$ solution dissolved in the electrolyte.

## Determination of $NO_2^-$

In all, 0.1 mL of mixture including 2 M HCl and 10 mg mL$^{-1}$ sulfanilamide and 5 mL diluted liquid products were first mixed. Then, 0.1 mL N-(1-Naphthyl) ethylenediamine dihydrochloride solution (10 mg mL$^{-1}$) was added to above mixed solution. The acquired solution was rested for 30 min under ambient conditions. Finally, the absorbance at the wavelength of 540 nm was recorded. The concentration-absorbance calibration curve was plotted using a series of concentrations known as standard $KNO_2$ solution dissolved in the electrolyte.

## Determination of $N_2H_4$

The presence of $N_2H_4$ in the electrolytes was determined utilizing the Watt-Chrisp method[55]. A color reagent was prepared by combining 100 mL ethanol, 2 g para(dimethylamino) benzaldehyde, and 12 mL concentrated HCl. Subsequently, 2 mL of the color reagent was introduced into 2 mL of the electrolyte. Following a 30-minute incubation period, the absorbance at the wavelength of 458 nm was recorded. A concentration-absorbance calibration curve was then constructed using a series of known standard $N_2H_4$ solution dissolved in the electrolyte.

## Isotope labeling experiments

A 0.5 M solution of K$^{15}$NO$_3$ (pH 1) served as the electrolyte in the cathode compartment. To facilitate quantitative determination, a certain amount of maleic acid (C$_4$H$_4$O$_4$) was dissoved in electrolyte solution with a concentration of 400 ppm. Subsequently, 50 μL of deuterium oxide (D$_2$O) was added to 0.5 mL of the aforementioned mixture for $^1$H NMR (400 MHz) detection[55]. $NH_3$ detection via NMR was also conducted in an electrolyte consisting of 0.1 M H$^{14}$NO$_3$/0.4 M K$^{14}$NO$_3$. The subsequent steps followed the previously outlined procedure. The peak area ratio of $NH_4^+$ to C$_4$H$_4$O$_4$ was recorded, and the concentration was determined based on standard calibration curves. These calibration curves were established by collecting peak area ratios of $NH_4^+$ at various concentrations of $^{14}$NH$_4^+$ and C$_4$H$_4$O$_4$.

## In-situ FTIR test

In-situ FTIR spectroscopy experiments were conducted using a modified electrochemical cell (see Supplementary Fig. 45) integrated into a Nicolet 6700 FTIR spectrometer equipped with a mercury-cadmium-telluride (MCT) detector cooled by liquid nitrogen. The working electrode, counter electrode, and reference electrode utilized were the self-supported catalysts electrode, Pt wire, and a Ag/AgCl electrode, respectively. Each spectrum was acquired with a time resolution of 30 seconds. Prior to the test, the initial state with an open circuit was scanned for background correction.

## FE and yield rate for $NO_3^-$RR

FEs and area-normalized yield rates of $NH_3$, $NO_2^-$, and $N_2H_4$ were determined using the following equations[55]:

$$FE\,(NH_3) = (8\,F \times C \times V \times n)/Q \tag{1}$$

$$Yield\,rate\,(NH_3) = (C \times V \times n)/(t \times A) \tag{2}$$

$$FE\,(NO_2^-) = (2\,F \times C \times V \times n)/Q \tag{3}$$

$$Yield\,rate\,(NO_2^-) = (C \times V \times n)/(t \times A) \tag{4}$$

Where F represents the Faraday constant (96485 C mol$^{-1}$), C denotes the measured concentration of $NH_3$, V signifies the volume of the electrolyte, Q represents the total charge passed through the electrode, n stands for the dilution factor, and A signifies the geometric area of the working electrode (0.5 × 0.5 cm$^2$).

## Computational details

Vienna Ab initio Simulation Package (VASP) was used to conduct spin-polarized density functional theory (DFT) calculations. The Perdew–Burke–Ernzerhof (PBE) functional was utilized for exchange-correlation treatment using the Generalized Gradient Approximation (GGA). The projector-augmented wave (PAW) method was employed to describe the core electrons, and the valence electronic states were expanded in plane-wave basis sets. Dispersion interactions were treated using the DFT-D3 method with Becke–Johnson (BJ) damping. A cutoff energy of 450 eV was set, and the force convergence criterion was set to 0.05 eV/Å. The optimization of the anatase-$TiO_2$(101) surface was carried out using a (3x3x1) k-point, while a Gamma point was adopted for the $TiO_2$/FePc surface optimization.

## Data availability

Data supporting the findings of this study are available from the corresponding author upon reasonable request.

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

## Acknowledgements

This work was supported by T23-713/22-R by RGC. We thank Dr. M. K. TSE from the Department of Chemistry of the City University of Hong Kong for the NMR measurements.

## Author contributions

R.Z., Y.G., and C.Z. conceptualized the project. Y.G. and C.Z. supervised the project. R.Z., C.L., P.L., Y.H., G.L., and Z.H. planned and performed the catalyst synthesis, conducted the electrocatalytic tests, collected, and analyzed the data. C.P. did the theoretical calculations. C.L. and H.C. collected and analyzed the NMR data. Y.W. and S.Z. helped with synthesis of the catalysts and collected the data. R.Z., Y.G., and C.Z. wrote the manuscript. All authors discussed the results and commented on the manuscript.

## Competing interests

The authors declare no competing interests.
