## [Peer Review File · Nature Communications]

REVIEWER COMMENTS

Reviewer #1 (Remarks to the Author):

This paper reported the application of Fe phthalocyanine/TiO₂ (FePc/TiO₂) for energy-efficient NO₃-RR in strong acid conditions with high NH₃ yield rate of 17.4 mg h⁻¹ cm⁻² and FE of 90.6%. The authors further demonstrated an alkaline-acid hybrid Zn-nitrate battery (AAHZNB) with good performance, a high open-circuit voltage of 1.99 V with a power density of 91.4 mW/cm². The major issue with this study is that, although FePc molecules and TiO₂ have good chemical stability in acids, as the authors pointed out themselves, the FePc and TiO₂ substrates form a composite catalyst only through van der Waals forces, and this weak interaction is intrinsically unstable to construct a practical electrocatalysts. Indeed, the stability duration of only three hours is too short, and Fig. 3e shows a clear reduction in the Fe-N bond, indicating that FePc got detached from the TiO₂ under operando conditions. This severe and intrinsic stability issue, along with other technical issues I listed below, make this work unsuitable for publication in this journal.

Other issues:

1. Both Figs. 1a and 1c mention that more proton sources can promote the reduction of NO₃⁻, which can lead to a lower overpotential. However, it is wired to notice that the onset potentials in Figs. 1b and 1d for the strong base condition (pH=13) are comparable to those for the strong acid condition (pH=1). Why does the alkaline condition also significantly improve the performance of NO₃RR (compared to pH=7)? This seems to be different from the previous conclusion, detailed discussion is needed.
2. For control samples, the authors only tested the properties of TiO₂ and did not investigate the small molecule FePc. Please complete the performance of FePc and discuss whether FePc also has catalytic activity for NO₃RR.
3. Fig. 2f has only used LSVs to elaborate the effect of NO₃⁻ concentration modulation on performance, which is not rigorous. The authors are suggested to complete the performance data of FePc/TiO₂ at different concentrations and, if possible, to taper the concentration to saturation (such as 2 mol-1) in order to further evaluate the catalytic activity of this catalyst under extreme environments.
4. In Fig. 2i, the authors only list the NO₃RR performance of the material compounded with TiO₂, which is not representative of the state-of-the-art NO₃RR investigations and seems to be a deliberate avoidance of the discussion of the insufficient performance, even though the reaction in this work occurs in strong acidic conditions with presumable more efficient proton supply. To the best of my knowledge, there are many published papers with better ammonia production rates (Nat. Commun. 2022, 13, 1129; JACS 2020, 142, 7036; Nat. Commun. 2021, 12, 2870; ACS Nano 2022, 16, 1072; ACS Catal. 2021, 11, 15135; ACS Nano 2022, 16, 4795, etc.). Please make more thorough evaluation based on the recently published catalyst properties regarding NO₃RR. It is also desirable to include the mass activity and to investigate the effect of catalyst loading on performance to further explore the application potential of this system.

5. The authors show a series of NO₃⁻ reducing intermediates in the test process through Figs. 3g and 3h, but these do not directly prove that Fe is involved in the catalysis in this process. Is there spectral evidence for Fe adsorption of NO₃⁻ or detection of Fe-NO₃⁻ intermediates? How to prove that Fe is the active center?
6. The authors should to complete the DFT results for FePc molecules and discuss the data.
7. In Fig. 5 the authors discuss that the acid-base mixed Zn-NO₃⁻ battery seems to perform better in this system than the battery in a neutral/alkaline environment. However, what are the essential advantages that the acidic NO₃RR can actually bring to the use of Zn-NO₃⁻ battery? In other words, if other NO₃RR systems with higher NH₃ production rate than the FePc-TiO₂ were used, even in alkaline/neutral conditions, will a better Zn-NO_x cell performance be achieved?
8. In addition, it seems that the acid-base hybrid battery with two separate poles also requires the use of acid-base membrane, which makes the battery rather complicated with higher cost and lower durability.

Reviewer #3 (Remarks to the Author):

The authors report a high ammonia yield rate of 17.4 mg h⁻¹ cm⁻² and a faradaic efficiency of 90.6% over the FePc/TiO₂ catalyst. An alkaline-acid hybrid Zn-NO₃⁻ battery (AAHZNB) for high-efficient NH₃ production was also developed. This work shows the potential of acidic NO₃-RR for NH₃ electrosynthesis and develops a high-performance hybrid Zn-NO₃⁻ battery and further broadens the field of fuel cells by using NO₃⁻ as fuel. However, there are several issues that need to be addressed.

Comments

- 1) There have been various studies in the literature which report high ammonia yield in acidic medium. How is this work novel as compared to these studies?
<https://onlinelibrary.wiley.com/doi/full/10.1002/anie.202305246>,
<https://pubmed.ncbi.nlm.nih.gov/16228644/>,
<https://www.sciencedirect.com/science/article/pii/S0013468616327050>
- 2) In line 37, where you discuss the Zn-NO₃⁻ batteries, please provide some literature review about the already existing ones and how this one holds more advantages than them.
- 3) In Figure 2, please include ammonia current densities as well.

4) NMR was done to confirm the presence of NH₃ for isotopic labelling cases. Please perform NMR for some of the N¹⁴ cases as well as UV-Vis spectrometry is subjected to false positives especially for NH₃.

5) In line 111 authors mention that TiO₂ enables more ammonia formation at lower pH. Please explain the mechanism and the process fundamentally in order to provide sufficient reasoning for this.

6) In line 142, authors mention that they chose 0.5M nitrate concentration for their studies. Please study the effect of nitrate concentration and include a figure pertaining to that.

7) In line 154, authors claim that hydrazine wasn't formed. Please explain how hydrazine was quantified in this study.

In the manuscript entitled "Electrochemical nitrate reduction in strong acid enables high-efficiency ammonia synthesis and high-voltage pollutant-based fuel cells," Zhang et al. reported the high-efficiency nitrate reduction to ammonia in acid media. It is interesting as most current work were carried out under neutral or alkaline conditions and the performance of FePc/TiO₂ for acidic nitrate reduction is impressive. Taking advantages of the acidic nitrate reduction in high potential and large current, the multifunctional Zn-nitrate battery and hydrazine-nitrate fuel cell have been developed for ammonia synthesis, pollutant removal and power generation. Overall, this is an interesting work that may provide new insights into the electrochemical nitrate reduction to ammonia and important concept for designing multifunctional aqueous battery related to catalysis, energy, and environment. I think it could be published after minor revisions. Below are my questions and comments.

1. It is suggested to provide more representative keywords in this work, such as acidic conditions, hydrazine-nitrate fuel cells and so on.
2. As little work has been reported about the acidic nitrate reduction to ammonia, I think that it should elaborate more on the advantages of acidic catalysis in the introduction part. For example, When carried out in acidic media, the NH₄⁺ ions formed are in the aqueous phase, eliminating the need of post-treatment and allowing direct application as fertilizers to be absorbed by the plants. In addition, what is the aftertreatment of acidic electrolytes? And will it cause acid any pollution to the environment?
3. The concentration of nitrate in the electrolyte is 0.5 M, which is high than many reported works. I wonder know how about the performance of the FePc/TiO₂ for nitrate reduction in lower concentrations.
4. In the Figure 6c, UV-Vis spectra were provided to identify the short-chain S₂²⁻ oxidation product of S₂⁻, is there any methods to collect these product for further analysis?
5. It is mentioned a stirring was kept for the electrochemical tests to minimize the mass transfer limitation. If so, how about the power density of Zn-nitrate battery working in the flow mode compared in the stirring condition?
6. Please supplement the error bars in Supplementary Figure 13. In addition, the colors in Supplementary Figure 13a and b should be readjusted.
7. More detailed description of the test should be provided in this work, for example, the distance between the two electrodes during the battery test; the area of the membrane of the counter electrode; the detailed of the battery test and in-situ FTIR.
8. There are some minor issues that should be addressed, for example, Page 19, line 278, replace "vis" by "Vis"; Page 12, line 163, replace "Table S1" by "Supplementary Table 1; replace "Table S2" by "Supplementary Table 2; Supplementary Table 3 should be added together with Supplementary Table 2.

Response Letter

Reviewer 1

This paper reported the application of Fe phthalocyanine/TiO₂ (FePc/TiO₂) for energy-efficient NO₃⁻RR in strong acid conditions with high NH₃ yield rate of 17.4 mg h⁻¹ cm⁻² and FE of 90.6%. The authors further demonstrated an alkaline-acid hybrid Zn-nitrate battery (AAHZNB) with good performance, a high open-circuit voltage of 1.99 V with a power density of 91.4 mW/cm². The major issue with this study is that, although FePc molecules and TiO₂ have good chemical stability in acids, as the authors pointed out themselves, the FePc and TiO₂ substrates form a composite catalyst only through van der Waals forces, and this weak interaction is intrinsically unstable to construct a practical electrocatalysts. Indeed, the stability duration of only three hours is too short, and Fig. 3e shows a clear reduction in the Fe-N bond, indicating that FePc got detached from the TiO₂ under operando conditions. This severe and intrinsic stability issue, along with other technical issues I listed below, make this work unsuitable for publication in this journal.

Response: Thanks for your constructive comments to help us improve the manuscript.

In fact, it is mentioned that **a clear chemical bond (~2.28 Å in length) is formed between the Fe site and the lattice oxygen on the TiO₂ surface within the FePc/TiO₂ composite** in the original manuscript. Conversely, the remaining portion of the FePc primarily exhibits van der Waals interactions with the TiO₂ substrate. Please don't misunderstand. We have improve our expression the revised version.

Further analysis, involving charge density difference assessments and density of states examinations, substantiates a considerable charge transfer (0.12 e) and robust Fe-O 3d-2p orbital overlap below the Fermi level between FePc and TiO₂ (Fig. R1), thus confirming a genuine chemical interaction rather than merely van der Waals interactions.

To assess the stability of the Fe-O chemical bond during NO₃⁻ reduction, we performed relative energy calculations, which can quantifiably measure the interactions between FePc and TiO₂. Our findings reveal an interaction energy of -2.55 eV between FePc and TiO₂, with this value remaining consistently close to -2.5 eV even after the adsorption of intermediate species on the Fe site (Fig. R2,3). This consistency demonstrates the enduring and robust nature of the chemical interactions with the FePc/TiO₂ composite.

To investigate the long-term stability, we extended the electrolysis of FePc/TiO₂-2 to a duration of ~24 hours from the original 3 hours in 60-mL electrolyte of 0.5 M NO₃⁻ (pH 1). As shown in Fig. R4, the current density exhibits a slight decrease during the initial two hours, followed by a sustained and nearly constant trend in the subsequent time period. The NH₃ FE recorded at different times remains almost stable and only decreases slightly in the final 6 hours from 86.6% to 81.9%. These results indicate the good electrochemical stability of FePc/TiO₂-2.

X-ray Photoelectron Spectroscopy (XPS) measurement is highly influenced by the surface state of the material (J. Vav. Sci. Technol. A 2020, 38, 063204). When contrasted with its initial dry state, prolonged immersion of the catalyst in the aqueous solution

may influence the intensity of element signals detected by XPS, particularly regarding nitrogen element with light atomic weight. Additionally, it is essential to note that XPS is a semi-quantitative technique. Hence, deriving the precise elements' content solely based on its peak signal is not a scientifically justifiable approach. To explore whether FePc got detached from the TiO₂ during the electrolysis, which may be related to the slight loss of NO₃⁻RR activity of FePc/TiO₂ during long-term stability test, Fourier transform infrared spectroscopy (FTIR), energy dispersive X-ray spectroscopy (EDS) and scanning electron microscope (SEM) analyses are carried out, revealing the presence of FePc and TiO₂ in the tested electrode (Fig. R5). Then, we collected the solution after electrolysis using FePc/TiO₂-2 and conducted inductively coupled plasma-mass spectrometry (ICP-MS) measurement for both Fe and Ti elements. As shown in Fig. R6, only 5.17 weight percent (wt%) of Fe dissolved in the first 4 hours and this figure increases to about 20 wt% after 24-hour electrolysis. It should be noted that the Ti element is also detected in the solution, suggesting the inevitably partial dissolution of the overall FePc/TiO₂-2 electrode in the acidic conditions after long-term electrolysis, which may be associated with the slight loss of electrocatalytic NO₃⁻RR activity. It also proves that FePc is not detached from TiO₂ catalyst electrode. This corrosion phenomena in strong acidic media are actually common in reported literature (ACS Catal. 2015, 5, 6503–6508; Chem. Soc. Rev. 2017, 46, 1933-1954) but FePc/TiO₂-2 still is active for acidic NO₃⁻RR. Furthermore, the long-term stability of the highly active FePc/TiO₂-2 electrode could be enhanced by rational optimization of intercomponent interaction, phase composition, and microstructure of the electrode.

These represent the focal points of our ongoing experimental endeavors.

Based on the above analyses, we believe that the FePc/TiO₂ in this work shows good stability in electrochemical activity and structure and can be considered as a practical electrocatalyst for NO₃⁻RR applications.

Fig. R1. (a) Charge density difference for FePc/TiO₂ system. (b) Density of states of Fe-O orbitals in FePc/TiO₂.

Fig. R2. Atomic configurations of different N-containing intermediates adsorbed on FePc/TiO₂-2 for NO₃⁻RR.

Fig. R3. Interaction energy between FePc and TiO₂ in the composite catalyst before and after adsorbed with different N-containing intermediates for NO₃⁻RR.

Fig. R4. Chronoamperometry curve of FePc/TiO₂-2 at -0.25 V in 0.5 M NO₃⁻ solution (pH 1) for NO₃⁻RR and NH₃ FE at different times.

Fig. R5. (a) FTIR spectrum and (b) SEM image and corresponding EDS mappings of FePc/TiO₂-2 after stability test.

Fig. R6. Dissolved mass ratio of Fe and Ti in the solution at different electrolysis time.

Please check the highlighted discussion on page 13–15, 17 in the revised manuscript and the newly added Fig. 25, 27, 28, 32–34, 37 in the revised Supplementary information.

Other issues:

1. Both Figs. 1a and 1c mention that more proton sources can promote the reduction of NO_3^- , which can lead to a lower overpotential. However, it is wired to notice that the onset potentials in Figs. 1b and 1d for the strong base condition (pH=13) are comparable to those for the strong acid condition (pH=1). Why does the alkaline condition also significantly improve the performance of NO_3^- RR (compared to pH=7)? This seems to be different from the previous conclusion, detailed discussion is needed.

Response: Thank you for your constructive suggestions. We are sorry for our oversight. We recollected the linear sweep voltammetry (LSV) curves of the TiO_2 in 0.5 M NO_3^- at different pH values of 1, 7 and 13. As shown in Fig. R7a, the onset potential for acidic NO_3^- RR is larger than that for neutral medium, followed by alkaline medium, and the current density in acid is always higher than that for neutral and alkaline media at the same potential. However, with the potential being more negative, the current density

for alkaline medium surpasses that for neutral media due to the slow reaction kinetics in neutral medium, as evidenced by the more sluggish kinetics shown in Tafel slopes and Nyquist plots (Fig. R7b,c). It is also clear that the acidic and alkaline electrolytes (pH 1 and pH 13) have lower ohmic resistance losses because of higher ionic conductivity compared to the neutral medium (*Mater. Horiz.* 2016, 3, 169–173). We then collected the FE toward NH_3 formation at different potentials in three media after electrolysis (Fig. R7d). At pH=1, TiO_2 shows high NH_3 FEs (from 66% to 78.5%) at the potentials in the range of -0.05 V and -0.55 V. However, the NH_3 FEs in the neutral/alkaline medias are very low (around 20%) at the first potential, increasing with the potential being more negative. The highest NH_3 FE in acidic conditions is 78.5% at -0.25 V while it is 74.3% at -0.45 V for neutral media and 68.8% at -0.45 V for alkaline media. The NH_3 partial current densities at different potentials and pH values of TiO_2 are shown in Fig. R7e. It is clear that TiO_2 consistently exhibits a higher NH_3 yield rate at pH 1 than that at pH 7 and pH 13 at the same applied potential, suggesting that acidic media enables a higher NH_3 production yield.

In order to investigate the pH-dependent influences on the reaction pathways, we have examined the evolution of free energy plots at pH values of 1, 7, and 13. It is worth noting that the Gibbs free energy has been adjusted in accordance with pH corrections as detailed in a prior report (*ACS Catal.* 2021, 11, 14417–14427). Our analysis of the pH-dependent influences on the free energies of reaction intermediates is elucidated through the role of H^+ . Whether we consider the deoxygenation of $^*\text{NO}_x$ or the hydrogenation of $^*\text{NH}_y$, it becomes evident that NO_3^- RR exhibits favorable energetics when H^+ ions are readily available within an acidic medium. This trend is substantiated by the energy evolution diagram for NO_3^- RR presented in Fig. R8. As the pH increases to 7 and 13, a concomitant increase in the free energies for NO_3^- RR is observed due to

the sluggish kinetics of H^+ produced from additional water dissociation. The foregoing analysis further suggests that an acidic environment may be more conducive to NO_3^- RR for TiO_2 . All these results demonstrate that acidic media enables faster hydrogenation kinetics for NO_3^- RR and more energy-efficient NH_3 synthesis for TiO_2 compared to both neutral and alkaline media.

Fig. R7. (a) LSV curves, (b) Tafel plots and (c) Nyquist plots of TiO_2 for NO_3^- RR with $0.5 NO_3^-$ in the electrolyte at different pHs. (d) NH_3 FE and (e) NH_3 partial current densities (j_{NH_3}) of TiO_2 for NO_3^- RR with $0.5 NO_3^-$ in the electrolyte at different pHs and different potentials.

Fig. R8. Gibbs free energies for NO_3^- RR on TiO_2 at $pH = 1, 7$ and 13 .

Please check the modified Fig. 1b–d, the highlighted discussion on page 7–9 in

the revised manuscript and the newly added Fig. 1, 9, 10 in the revised Supplementary information.

2. For control samples, the authors only tested the properties of TiO₂ and did not investigate the small molecule FePc. Please complete the performance of FePc and discuss whether FePc also has catalytic activity for NO₃⁻RR.

Response: Thank you for your valuable comment. Following your suggestion, we have supplemented the performance of FePc for NO₃⁻RR. From the LSV measurement in Fig. 2d and 2e in the original manuscript, the obvious increase in current density after replacing ClO₄⁻ with NO₃⁻ indicates that the FePc has catalytic activity for NO₃⁻RR. We also performed electrolysis at different potentials for a quantitative determination of NO₃⁻ conversion to NH₃. As shown in Fig. R9, FePc shows an increased NH₃ formation rate with the potential being more negative in the range of -0.05 V to -0.75 V vs. RHE with 0.5 NO₃⁻ at pH 1. In addition, the FE for NH₃ reaches the maximum value of 76.0% at -0.45 V, lower than that for FePc/TiO₂-2 All these results prove that FePc shows catalytic activity for NO₃⁻RR.

Fig. R9. (a) NH₃ yield and (b) NH₃ FE of FePc in 0.5 NO₃⁻ at pH = 1.

Please check the highlighted part on page 12 in the revised manuscript and the newly added Fig. 22 in revised Supplementary information.

3. Fig. 2f has only used LSVs to elaborate the effect of NO_3^- concentration modulation on performance, which is not rigorous. The authors are suggested to complete the performance data of FePc/TiO₂ at different concentrations and, if possible, to taper the concentration to saturation (such as 2 mol⁻¹) in order to further evaluate the catalytic activity of this catalyst under extreme environments.

Response: Thanks for your insightful comment. We are sorry for not completing the NO_3^- RR performance of the FePc/TiO₂-2 at different concentrations. Following your suggestion, we performed the electrolysis at different potentials and different concentrations. The obtained NH_3 yield and corresponding FE for a better comparison are shown in Fig. R10a. The NH_3 formation rate of FePc/TiO₂-2 shows an increased trend with the increased NO_3^- concentrations from 0.1 M to 2 M in the solution at all potentials. When NO_3^- concentration is up to 2 M, NH_3 formation rate still significantly increases and reaches 22.5 mg h⁻¹ cm⁻², indicating that the FePc/TiO₂ can work under extreme environments. Besides, the calculated NH_3 FE also shows similar increased trends with increased NO_3^- concentration at each potential (Fig. R10b), suggesting that the high NO_3^- concentration fascinates the NH_3 formation. Specifically, the maximum NH_3 FE values for 0.1 M, 0.2 M, 0.3 M, 0.4 M and 0.5 M NO_3^- electrolytes using FePc/TiO₂-2 are determined as 77.0%, 79.2%, 83.2%, 87.9% and 90.6%, respectively. The NH_3 FE in 2 M NO_3^- is 92.7%, which is close to that in 0.5 M NO_3^- electrolytes. Therefore, we chose 0.5 M for NO_3^- RR in the next explorations.

Fig. R10. (a) NH₃ yield and (b) NH₃ FE of FePc/TiO₂-2 at different potentials and different NO₃⁻ concentrations of 0.1 M, 0.2 M, 0.3 M, 0.4 M, 0.5 M and 2 M.

Please check the highlighted discussion on page 11 in the revised manuscript and the newly added Fig. 15, 16 in the revised Supplementary information.

4. In Fig. 2i, the authors only list the NO₃RR performance of the material compounded with TiO₂, which is not representative of the state-of-the-art NO₃RR investigations and seems to be a deliberate avoidance of the discussion of the insufficient performance, even though the reaction in this work occurs in strong acidic conditions with presumable more efficient proton supply. To the best of my knowledge, there are many published papers with better ammonia production rates (Nat. Commun. 2022, 13, 1129; JACS 2020, 142, 7036; Nat. Commun. 2021, 12, 2870; ACS Nano 2022, 16, 1072; ACS Catal. 2021, 11, 15135; ACS Nano 2022, 16, 4795, etc.). Please make more thorough evaluation based on the recently published catalyst properties regarding NO₃RR. It is also desirable to include the mass activity and to investigate the effect of catalyst

loading on performance to further explore the application potential of this system.

Response: Thank you for your valuable suggestions. We tend to select the reported supported TiO₂-based electrocatalysts to compare with our FePc/TiO₂-2 in terms of NH₃ synthesis performance for fairness. According to your comment, we also supplemented more performance comparisons of the state-of-the-art NO₃⁻RR electrocatalysts, as shown in Table R1. It is clear that the FePc/TiO₂-2 still shows a higher NH₃ synthesis rate than all the reported TiO₂-based electrocatalysts in neutral and alkaline media. We have to acknowledge that there is still a gap in performance between FePc/TiO₂-2 and the state-of-the-art NO₃⁻RR electrocatalysts like Ru-dispersed Cu nanowire (Nat. Nanotechnol. 2022, 17, 759–767) and Ru nanoclusters (J. Am. Chem. Soc. 2020, 142, 7036–7046), which performed in strong alkaline media with high NO₃⁻ concentration up to 1 M. However, it can be confirmed that the precious-metal-free FePc/TiO₂-2 shows higher NH₃ synthesis performance than RuCu (Adv. Energy Mater. 2022, 12, 2103916) and Fe₂M-trinuclear-cluster metal-organic frameworks (Angew. Chem. Int. Ed. 2023, 62 202305246) catalysts recently reported in acid.

We further investigate the mass activity of FePc/TiO₂ with different FePc loading masses. Here the loading mass of FePc/TiO₂-1, FePc/TiO₂-2 and FePc/TiO₂-3 are estimated as 0.512 mg cm⁻², 0.526 mg cm⁻², 0.531 mg cm⁻², respectively. FePc/TiO₂-2 shows a higher geometrical area and mass normalized NH₃ yield than FePc/TiO₂-1 and FePc/TiO₂-3 (Fig. R11a,b), revealing that higher or lower FePc mass loading would lead to the decreased NH₃ yield. Besides, it shows similar peak FE values for NH₃ formation with 88.7%, 90.6% and 87.4% for FePc/TiO₂-1, FePc/TiO₂-2 and FePc/TiO₂-3 (Fig. R11c), respectively, suggesting that the catalyst loading shows little impact on the NH₃ FE of NO₃⁻RR.

Table R1. Comparison of the recently reported electrocatalysts for NO_x⁻ reduction in different electrolytes.

Catalysts	Electrolyte	Maximum NH ₃ FE (%)	NH ₃ yield (mg h ⁻¹ cm ⁻²)	NH ₃ yield (mg h ⁻¹ mg _{cat.} ⁻²)	Ref.
TiO _{2-x}	0.5 M Na ₂ SO ₄ /500 ppm NO ₃ ⁻ -N	85	0.3875	0.765	1
Pd/TiO ₂	0.25 M LiNO ₃ /5 M LiCl	92.1	1.12	0.112	2
Ru NCs/TiO ₂	100 ppm NO ₃ ⁻ /0.05 M Na ₂ SO ₄	90	10.2	26.8	3
Cu/Fe-TiO ₂	50 ppm NaNO ₃ /0.5 M Na ₂ SO ₄	91.2	8.5974	8.5974	4
Co@TiO ₂	0.1 M PBS/0.1 M NO ₃ ⁻	96.7	13.6	11.33	5
CoP@TiO ₂	0.1 M NaOH/0.1 M NO ₃ ⁻	96.6	8.4966	-	6
Co ₃ O ₄ @TiO ₂	0.1 M NaOH/0.1 M NO ₃ ⁻	93.1	14.875	9.917	7
Ru ₁ /TiO _x	1 M KOH/1 M NaNO ₃	87.3%	9.42	-	8
Ni-TiO ₂	0.1 M NaOH/0.1 M NO ₂ ⁻	94.89	6.46459	-	9
Ag@TiO ₂	0.1 M NaOH/0.1 M NO ₂ ⁻	96.4	8.7431	-	10
V-TiO ₂	0.1 M NaOH/0.1 M NO ₂ ⁻	93.2	9.1936	-	11
Ni@TiO ₂	0.1 M NaOH/0.1 M NO ₂ ⁻	98.5	9.6679	-	12
NiS ₂ @TiO ₂	0.1 M NaOH/0.1 M NO ₂ ⁻	92.1	10.0623	-	13
CuCoSP	0.1 M KOH/0.1 M NO ₃ ⁻	93.3	19.89	-	14
Ru nanocluster	1 M KOH/1 M KNO ₃	100	19.89	94.52	15
Fe SAC	0.5 M KNO ₃ /0.1 M K ₂ SO ₄	75	7.82	20	16
Fe-PPy SACs	0.1 M KOH/0.1 M NO ₃ ⁻	100	2.75	11.45	17
Fe-N/P-C	0.1 M KOH/0.1 M NO ₃ ⁻	90.3	8.99	17.98	18
Fe-cyano-R NSs	1 M KOH/0.1 M KNO ₃	90.4	21.1	42.1	19
CoO _x	0.1 M KOH/0.1 M KNO ₃	93.4	6.592	82.4	20
BiOCl	1 M KOH/0.5 M KNO ₃	90.6	23.715	46.5	21
Ru-CuW	1 M KOH/0.1 M NO ₃ ⁻	95.6	76.512	191.28	22
CuNi alloys	1 M NaOH/44.3 g L ⁻¹ NO ₃ ⁻	97.03	94.57	-	23
0.6W-O-CoP	1 M KOH/0.1 M NO ₃ ⁻	80.92	88.9	31.6	24
CoP NAs	1 M KOH/1 M NO ₃ ⁻	100	16.25	9.67	25
RuFe NFs	0.1 M NaNO ₃ /0.5 M Na ₂ SO ₄	85.1	7.74	38.68	26
Ru/β-Co(OH) ₂	1 M KOH/1 M KNO ₃	98.78	39.1	24.12	27
Fe ₂ M-MOF	0.05 M H ₂ SO ₄ /50 g L ⁻¹ KNO ₃	90.55	10.65	20.65	28
PA-RhCu	0.1 M HClO ₄ /0.05M NO ₃ ⁻	93.7	0.24	2.4	29
FePc/TiO₂-2	0.1 M HNO₃/0.4 M KNO₃	90.6	17.4	33.08	This work

Fig. R11. NH₃ yield normalized by (a) geometrical area and (b) catalyst mass and (c) NH₃ FE of FePc/TiO₂-1, FePc/TiO₂-2 and FePc/TiO₂-3 at different potentials.

Please check the highlighted part on page 12, 13 in the revised manuscript and the newly added Fig. 23, Table 1 in the revised supplementary information.

5. The authors show a series of NO₃⁻ reducing intermediates in the test process through Figs. 3g and 3h, but these do not directly prove that Fe is involved in the catalysis in this process. Is there spectral evidence for Fe adsorption of NO₃⁻ or detection of Fe-NO₃⁻ intermediates? How to prove that Fe is the active center?

Response: Thank you for your valuable comments. To identify the NO₃⁻ adsorbed on Fe site, FePc/TiO₂-2 was immersed in the electrolyte solution for 24 hours and then collected the XPS spectra before and after immersion for further analysis. As shown in Fig. R12a, the typical peak for NO₃⁻ appears in the spectra of FePc/TiO₂-2 after immersion. In addition, the peak for Fe 2p_{3/2} region shifts toward higher binding energy compared to that before the immersion (Fig. R12b), indicating the possible interaction between NO₃ and Fe centers after adsorption. In-situ FTIR is an important technique to detect the intermediates adsorbed on the surface of the materials. Fig. R12c shows the in-situ FTIR spectra of FePc at open circuit potential (OCP) and -0.25 V. Several peaks at 1100, 1153, 1254, 1374 and 1630 cm⁻¹ can be assigned to *NH₃, *NH₂OH, *NO₂, *NO₃ and *H₂O/*NO, respectively (Small, 2023, 19, 2207743; Chem. Eng. J. 2022, 433, 133680; Adv. Funct. Mater. 2023, 33, 2211537), indicative the presence of NO₃⁻

adsorption on FePc and the occurrence of the deoxygenation and hydrogenation during the reduction process. Metal centers are generally considered active sites for NO_3^- RR. We next performed theoretical calculations and found that NO_3^- is preferably adsorbed on the Fe site with a larger adsorption energy of 1.48 eV compared to Ti (Fig. R12d), implying that Fe is more likely to be the active sites for NO_3^- RR.

To further elucidate the role of the active site of FePc/TiO₂ in NO_3^- RR, we conducted a series of SCN^- intoxication experiments to block Fe species of FePc/TiO₂-2 because of the strong affinity of SCN^- with Fe species (ACS Energy Lett. 2021, 6, 379–386; Angew. Chem. Int. Ed. 2023, DOI: 10.1002/anie.202308044). It was found that the NO_3^- RR performance of FePc/TiO₂-2 decreased significantly in terms of current density, NH_3 FE, and NH_3 yield after adding KSCN (Fig. R13), indicating the active role of Fe center in electrochemical NO_3^- RR. Besides, the pure FePc also shows catalytic activity for electrochemical NO_3^- reduction to NH_3 with a maximum FE of 76.0% (Fig. R9). All these results indicate that Fe is more likely the active center for NO_3^- RR.

Fig. R12. XPS spectra of FePc/TiO₂-2 in the (a) N 1s region and (b) Fe 2p region before and after immersion in 0.5 M NO₃⁻. (c) In-situ FTIR spectra of FePc at OCP and -0.25 V in acidic conditions. (d) Adsorption energy of NO₃⁻ at Ti and Fe site of FePc/TiO₂.

Fig. R13. (a) LSV curves of FePc/TiO₂-2 in 0.5 M NO₃⁻ solution with and without adding 0.02 mM KSCN. (b) Changes in current density before and after KSCN injection in 0.5 M NO₃⁻ solution at -0.25 V. (c) Comparison in terms of NH₃ yield and NH₃ FE of FePc/TiO₂-2 between initial and after the injection of KSCN.

Please check the highlighted part on pages 16–18 in the revised manuscript and the newly added Fig. 29–31, 35 in the Supplementary information.

6. The authors should to complete the DFT results for FePc molecules and discuss the data.

Response: Thank you for the comment. According to your suggestions, we supplemented the DFT calculations for the FePc molecule. Fig. R14a,b shows the atomic configuration of N-containing intermediates adsorbed on the FePc molecule and the corresponding free energy profile for NO_3^- RR at each step. The NO_3^- can be efficiently adsorbed on Fe centers with an adsorption energy of -1.76 eV. Then, the energy profile shows a downhill trend in the next reduction steps (Fig. R14b). We find that the reduction of *NO to *NOH is the rate-determining step for FePc during NO_3^- RR. It means that NO is an important reaction intermediate, which agrees with our in-situ FTIR spectra observations of FePc. In addition, Fig. R14c shows the Gibbs free energy for hydrogen evolution reaction (HER) on FePc with a large value of 0.38 eV, indicating the poor competing HER activity for FePc.

Fig. R14. (a) Atomic configurations of N-containing intermediates adsorbed on FePc. Free energy diagram for (b) NO_3^- reduction and (c) HER on the FePc (Inserted is the

adsorption configuration of H* on FePc).

Please check the highlighted part on page 19 in revised manuscript and newly added Fig. 38 in the revised supplementary information.

7. In Fig. 5 the authors discuss that the acid-base mixed Zn-NO₃⁻ battery seems to perform better in this system than the battery in a neutral/alkaline environment. However, what are the essential advantages that the acidic NO₃⁻RR can actually bring to the use of Zn-NO₃⁻ battery? In other words, if other NO₃⁻RR systems with higher NH₃ production rate than the FePc-TiO₂ were used, even in alkaline/neutral conditions, will a better Zn-NO_x⁻ cell performance be achieved?

Response: Thank you for the valuable comment. The power density (*P*) of Zn-NO₃⁻ battery was determined by $P = I \times V$, where *I* and *V* are the discharge current density and voltage, respectively. It is expected that a more positive potential and a higher catalytic current density for NO₃⁻RR at the cathode generally contribute to a higher output power of the Zn-NO₃⁻ battery. Therefore, we believe that there are two essential advantages that acidic NO₃⁻RR can actually bring to the use of Zn-NO₃⁻ battery in this work. The first is that the current density of NO₃⁻RR using FePc/TiO₂ increases more rapidly than that in neutral and alkaline media. The second is that the acidic medium endows a higher open-circuit voltage (OCV) of 1.99 V for the Zn-NO₃⁻ battery, larger than all of the reported Zn-NO₃⁻ batteries (Table R2). The potential of NO₃⁻ reduction versus standard hydrogen electrode (SHE) at different pH values are shown as below (Curr. Opin. Electroche. 2021, 28, 100721):

According to above equations, the OCV of alkaline-acid Zn-NO₃⁻ battery is about 1.58 times higher than that for all-alkaline Zn-NO₃⁻ battery, conducive to high power density of alkaline-acid Zn-NO₃⁻ battery. Such high OCV is ascribed to the intrinsic potential difference of Zn oxidation and NO₃⁻RR as well as the pH difference between the two chambers separated by a bipolar membrane, which can efficiently avoid the direct neutralization of acid and alkali. Fig. R15 shows LSV curves of anodic Zn oxidation of the Zn plate and the cathodic NO₃⁻RR using FePc/TiO₂ in different media at potentials versus Ag/AgCl reference electrode. It is clear that the acidic NO₃⁻RR shows a larger potential gap than that for neutral and alkaline media. Therefore, the onset discharging voltage of our Zn-NO₃⁻ battery is larger than 1 V and the peak power density reaches up to 91.4 mW cm⁻², exceeding all reported values of Zn-NO₃⁻ batteries up to now.

It is hard to say that a better Zn-NO_x⁻ cell performance can be developed with an efficient catalyst under neutral/alkaline media with a higher NH₃ production rate or higher catalytic current density than our FePc/TiO₂-2. In terms of NH₃ yield, we have to acknowledge that the recently reported CuNi alloy shows a higher NH₃ yield than FePc/TiO₂ (Energy Environ. Sci. 2023, 16, 2991). The CuNi alloys show ampere-level electrocatalytic NO₃⁻-to-NH₃ reduction activities. However, its power density of 70.7 mW cm⁻² is still lower than that for FePc/TiO₂. It indicates that the high voltage induced by acidic NO₃⁻RR endows great advantage for Zn-NO₃⁻ battery. On the other hand, if the catalytic current density or NH₃ yield rate can be further improved by rationally designing ultrahigh efficient electrocatalysts, we believe that a higher-performance Zn-NO₃⁻ battery will be developed in the near future. We thank the reviewer for this comment to inspire us the next experimental direction.

Table R2. Summary of the reported Zn-NO₃⁻ batteries. OCV is highlighted to show the

highest value achieved by our alkaline-acid Zn-NO₃⁻ battery.

Catholyte	Anolyte	Cathode	OCV (V)	NH ₃ yield (mg h ⁻¹ cm ⁻²)	P (mW cm ⁻²)	Membrane
0.1 M HNO₃ + 0.4 M KNO₃	6 M KOH	FePc/TiO₂-2 (This work)	1.99	12.3	91.4	Bipolar
0.25 M LiNO ₃ + 5 M LiCl	5 M KOH	Pd/TiO ₂	0.81	0.54	0.87	Bipolar
0.2 M K ₂ SO ₄ + 0.05 M KNO ₃	1 M KOH	Fe/Ni ₂ P	1.22	0.38	3.25	Bipolar
1 M KOH + 0.05 M NO ₃ ⁻	5 M KOH	MP-Cu	1.27	1.292	7.56	Nafion 117
0.5 M K ₂ SO ₄ + 200 ppm NO ₃ ⁻	-	Ni ₁ Cu-SAA	1.51	2.1	12.7	Nafion 117
1 M KOH + 1 M KNO ₃	1 M KOH + 0.02 M Zn(Ac) ₂	DM-Co	0.7	2.04	25.8	Anion membrane
1 M KOH + 1 M NaNO ₃	1 M KOH	Ru-25CV/NF	1.2	2.9	51.5	Bipolar
0.1 M NaOH + 0.1 M NO ₃ ⁻	6 M KOH	NiCo ₂ O ₄	1.3	0.82	3.94	Nafion 117
3 M KOH + 0.5 M NO ₃ ⁻	3 M KOH	Cu nanowire	0.943	2.125	14.1	Bipolar
0.1 M NaOH + 0.1 M NO ₃ ⁻	6 M KOH	Ir SAC Co ₃ O ₄	1.396	0.7633	5.6	Nafion 117
-	0.1 M NaNO ₃	Fe ₂ TiO ₅	1.5	0.7803	5.6	Nafion 1110
1 M KOH + 0.1 M NO ₃ ⁻	6 M KOH	NiCoBDC@HsGDY	1.47	1.125	3.66	Nafion 117
1 M KOH + 0.1 M NaNO ₃	1 M KOH	W-O-CoP	0.7	2.79	9.27	Alkaline membrane
0.5 M Na ₂ SO ₄ + 0.1 M NaNO ₃	1 M KOH + 0.02 M Zn(Ac) ₂	RuFe NF	1.37	-	9.5	Bipolar
3 M KOH + 0.5 M NO ₃ ⁻	3 M KOH + 0.1M Zn(Ac) ₂	NiRu ball-flower	1.39	-	10	Nafion 1110
1 M KOH + 0.1 M KNO ₃	6 M KOH	Ru/β-Co(OH) ₂	1.48	6.46	29.87	Nafion 212
3.5 M NaOH + 44.3 g L ⁻¹ NO ₃ ⁻	3.5 M NaOH	CuNi NPs/CF	0.94	18.1	70.7	Bipolar

Fig. R15. The LSV curves of anodic Zn oxidation of Zn plate and the cathodic NO₃⁻RR using FePc/TiO₂-2 at different pH values with potentials versus Ag/AgCl reference electrode.

Please check the highlight on page 20, 21 in revised manuscript and the newly added Fig. 39, modified Table 2 in the revised Supplementary information.

8. In addition, it seems that the acid-base hybrid battery with two separate poles also requires the use of acid-base membrane, which makes the battery rather complicated with higher cost and lower durability.

Response: Thank you for the comment. In order to make the battery system more stable, we have to use bipolar membrane with high efficiency and low crossovers. Such bipolar membrane has been widely used in reported Zn-NO₃⁻ battery systems, as shown in Table R2. It should be noted that the configuration of our acid-base hybrid battery is similar to the reported Zn-NO₃⁻ batteries except for separator membranes. As summarized in Table R2, Nafion 117 membranes are also commonly used in Zn-NO₃⁻ batteries. After checking the price of the bipolar membrane (Fumasep FBM-PK) used in this work and the Nafion 117 membrane in the Fuel Cell Store, we found that they are very close in price (0.47 \$ cm⁻²). However, we have to acknowledge that the price of the separator is still expensive compared to the porous polymer membrane and takes

up a large portion of the battery cost. It also motivates us to develop more efficient and low-cost separators in the future. Finally, we supplemented the stability of our acid-base hybrid battery. Fig. R16 shows the collected chronopotentiometry curves and the NH_3 FE at different times. The acid-base hybrid Zn-NO_3^- battery shows slightly decreased current density at the first 2 h at 0.9 V and maintains stable in the next 8 h. Besides, the NH_3 FE of the cathode changes from initial 84.7% to 81.8% after 10-hour electrolysis, indicative of the good stability of our batteries. Therefore, we believe that the developed acid-base hybrid Zn-NO_3^- battery is a highly efficient and stable system for simultaneous electricity supply and NH_3 synthesis.

Fig. R16. (a) General configuration of the reported Zn-NO_3^- battery systems. (b) Chronoamperometric curve of our acid-base hybrid battery for 10 hours and the NH_3 FE at different times.

Please check the highlight on page 22 in the revised manuscript and the newly added Fig. 42, Table 2 in the revised Supplementary information.

Reviewer 2:

In the manuscript entitled "Electrochemical nitrate reduction in strong acid enables high-efficiency ammonia synthesis and high-voltage pollutants-based fuel cells," Zhang et al. reported the high-efficiency nitrate reduction to ammonia in acid media. It is interesting as most current work were carried out under neutral or alkaline conditions

and the performance of FePc/TiO₂ for acidic nitrate reduction is impressive. Taking advantages of the acidic nitrate reduction in high potential and large current, the multifunctional Zn-nitrate battery and hydrazine-nitrate fuel cell have been developed for ammonia synthesis, pollutants removal and power generation. Overall, this is an interesting work that may provide new insights into the electrochemical nitrate reduction to ammonia and important concept for designing multifunctional aqueous battery related to catalysis, energy, and environment. I think it could be published after minor revisions. Below are my questions and comments.

Response: Thanks for recognizing our work.

1. It is suggested to provide more representative keywords in this work, such as acidic conditions, hydrazine-nitrate fuel cells and so on.

Response: Thank you for your valuable comments. Following your suggestion, we have provided more representative keywords in this paper, including NO₃⁻ reduction, acidic conditions, NH₃ electrosynthesis; Zn-NO₃⁻ battery, and N₂H₄-NO₃⁻ fuel cells.

Please check the highlighted text on page 3 in the revised manuscript.

2. As little work has been reported about the acidic nitrate reduction to ammonia, I think that it should elaborate more on the advantages of acidic catalysis in the introduction part. For example, when carried out in acidic media, the NH₄⁺ ions formed are in the aqueous phase, eliminating the need of post-treatment and allowing direct application as fertilizers to be absorbed by the plants. In addition, what is the aftertreatment of acidic electrolytes? And will it cause acid any pollution to the environment?

Response: Thank you for your insightful comments. As you suggested, we supplemented more potential advantages of the acidic nitrate reduction in the

introduction part. Direct NO_3^- reduction under strongly acidic conditions offers unique advantages compared with neutral/alkaline conditions. For example, the volatilization of NH_3 in neutral/alkaline electrolytes can be avoided and ammonium fertilizers/salts (i.e., NH_4NO_3 and $(\text{NH}_4)_2\text{SO}_4$) can be obtained directly to be absorbed by the plants (Nat. Nanotechnol. 2022, 17, 759–767; Curr. Opin. Electrochem. 2021, 28, 100721). As the reaction becomes more acidic, abundant protons are provided for continuous hydrogenation reactions of NO_3^- , guaranteeing the enhanced conversion rate toward NH_3 more energy-efficiently (J. Chem. Educ. 2018, 95, 84–87). The acids in the product can be easily neutralized/diluted with base/water and then directly applied as irrigation water with fertilizers, and therefore, the acidic electrolyte will not be released to act as a pollutant.

Please check the highlighted text on page 5 in the revised manuscript.

3. The concentration of nitrate in the electrolyte is 0.5 M, which is high than many reported works. I wonder know how about the performance of the FePc/TiO₂ for nitrate reduction in lower concentrations.

Response: Thanks for your insightful comment. Following your suggestion, we performed electrolysis in solution with lower NO_3^- concentrations, including 0.1 M, 0.2 M, 0.3 M and 0.4 M. The obtained NH_3 yield and corresponding FE are shown in Fig. R17a. It is clear that the NH_3 formation rate of FePc/TiO₂-2 shows an increased trend with the increased NO_3^- concentrations from 0.1 M to 0.5 M in the solution at all potentials. The calculated NH_3 FE also shows similar increased trends with increased NO_3^- concentration at each potential (Fig. R17b), suggesting that high NO_3^- concentration fascinates NH_3 formation. Specifically, the maximum NH_3 FE values for 0.1 M, 0.2 M, 0.3 M, 0.4 M and 0.5 M NO_3^- electrolytes using FePc/TiO₂-2 are

determined as 77.0%, 79.2%, 83.2%, 87.9% and 90.6%, respectively, suggesting that the FePc/TiO₂-2 are still active in lower NO₃⁻ concentrations. In addition, when NO₃⁻ concentration is up to 2 M, the NH₃ formation rate still significantly increases and reaches 22.5 mg h⁻¹ cm⁻² with a maximum NH₃ FE of 92.7%, implying that FePc/TiO₂-2 can work under extreme environments.

Fig. R17. (a) NH₃ yield and (b) NH₃ FE of FePc/TiO₂-2 at different potentials and different NO₃⁻ concentrations of 0.1 M, 0.2 M, 0.3 M, 0.4 M, 0.5 M and 2 M.

Please check the highlighted discussion on page 11 in the revised manuscript and the newly added Fig. 15, 16 in the revised Supplementary information.

4. In the Fig. 6c, UV-Vis. spectra were provided to identify the short-chain S₂²⁻ oxidation product of S²⁻, is there any methods to collect these products for further analysis?

Response: Thank you for your valuable suggestion. As verified by the UV-Vis. test results, polysulfides were generated from the oxidation product sulfur, further

combining with the sulfides in the electrolyte during the electrochemical S^{2-} oxidation process. The oxidation product can be efficiently obtained through the acid treatment of the electrolyte (Energy Environ. Sci. 2020, 13, 119–126). In detail, the concentrated sulfuric acid was added dropwise into the electrolyte solution until the pH to 1 in the ice bath, and then the yellow product was obtained by a centrifugal separating. Fig. R18 shows the X-ray diffraction pattern for the collected yellow powder, which corresponds to the elemental sulfur.

Fig. R18. X-ray diffraction pattern for the collected yellow powder.

Please check the highlighted text on page 23 in the revised manuscript, and the newly added Fig. 43 in Supplementary information.

5. It is mentioned a stirring was kept for the electrochemical tests to minimize the mass transfer limitation. If so, how about the power density of Zn-nitrate battery working in the flow mode compared in the stirring condition?

Response: To assemble a flow $Zn-NO_3^-$ battery, the electrolytes in the cathode and anode are circulated by a peristaltic pump (Kamoer, F01A-STP) with a flow rate of 25 mL min^{-1} . Fig. R19 exhibits the discharge curve of the flow $Zn-NO_3^-$ battery and corresponding power density. It is found that the flow $Zn-NO_3^-$ battery shows a peak

power density of 93.55 mW cm^{-2} , which is very close to that obtained from a static mode.

Fig. R19. The discharge curve of flow AAHZNB and the corresponding power density.

Please check the highlighted text on page 22 in the revised manuscript and the newly added Fig. 41 in Supplementary information.

6. Please supplement the error bars in Supplementary Fig. 13. In addition, the colors in Supplementary Fig. 13a and b should be readjusted.

Response: Thanks for your warm reminder. We are so sorry about our oversight. The error bars have been added, as shown in Fig. R20. In addition, the colors in Supplementary Fig. 13a and b have been adjusted accordingly.

Fig. R20. NO_2^- FE of (a) FePc/TiO₂-2 and (b) TiO₂ at pH1 with 0.5 M NO_3^- .

Please check the modified Fig. 19 in the revised Supplementary information.

7. More detailed description of the test should be provided in this work, for example, the distance between the two electrodes during the battery test; the area of the membrane of the counter electrode; the detailed of the battery test and in-situ FTIR.

Response: Thanks for your valuable comment. We are sorry for not having provided the enough details of the test. Following your suggestions, we supplemented more details of the experimental test in this work, including the electrochemical test, in situ FTIR test, battery assembly, etc.

Please find the highlighted parts on page 28–31 in revised manuscript.

8. There are some minor issues that should be addressed, for example, Page 19, line 278, replace "vis" by "Vis"; Page 12, line 163, replace "Table S1" by "Supplementary Table 1; replace "Table S2" by "Supplementary Table 2; Supplementary Table 3 should be added together with Supplementary Table 2.

Response: Thanks for your insightful comments. We are sorry for our oversight. As you suggested, we have carefully reviewed the manuscript to correct the previous mistakes and typos to increase the manuscript's science.

Reviewer 3

The authors report a high ammonia yield rate of $17.4 \text{ mg h}^{-1} \text{ cm}^{-2}$ and a faradaic efficiency of 90.6% over the FePc/TiO₂ catalyst. An alkaline-acid hybrid Zn-NO₃⁻ battery (AAHZNB) for high-efficient NH₃ production was also developed. This work shows the potential of acidic NO₃⁻RR for NH₃ electrosynthesis and develops a high-performance hybrid Zn-NO₃⁻ battery and further broadens the field of fuel cells by using NO₃⁻ as fuel. However, there are several issues that need to be addressed.

Response: Thank you for your recognition and valuable suggestions to help us improve

the manuscript. We have followed your comments and substantially revised our manuscript.

1) There have been various studies in the literature which report high ammonia yield in acidic medium. How is this work novel as compared to these studies?

<https://onlinelibrary.wiley.com/doi/full/10.1002/anie.202305246>

<https://pubmed.ncbi.nlm.nih.gov/16228644/>

<https://www.sciencedirect.com/science/article/pii/S0013468616327050>

Response: Thanks for your insightful comments. We have read these three works carefully. The first work reported a series of Fe₂M (M = Fe, Co, Ni, Zn) trinuclear cluster metal-organic frameworks (MOFs) for electrochemical NO₃⁻RR with NH₃ yield of 1.032675 mg h⁻¹ cm⁻² and FE of 90.55% (Angew. Chem. Int. Ed. 2023, 62, 202305246). The second work reported a ferrous sulfide (FeS) with higher chemical reduction activity for NO₃⁻/NO₂⁻ (Orig Life Evol Biosph 2005, 35, 299–312). The third work reported by Koper's group investigated the reaction mechanism and products on Cu (100) and Cu (111) in acid and alkaline. They concluded that NO₃⁻RR on Cu is pH-dependent and produces NO/NH₄⁺ in acid while generate NO₂⁻/NH₂OH in alkaline (Electrochim. Acta 2017, 227, 77–84).

We sincerely appreciate these reported works to enhance our comprehension of acidic NO₃⁻RR. After comparing above studies with our work, we would like to highlight the novelties of this work:

1. We first demonstrate the great potential of TiO₂ nanosheet with intrinsically poor hydrogen-evolution activity for selective and rapid NO₃⁻RR to NH₃ under strongly acidic conditions (pH=1). The NO₃⁻RR in an acidic medium exhibits a more rapid kinetics and low overpotential compared to those in neutral and alkaline media.

2. Hybridized with iron phthalocyanine (FePc), the resulting FePc/TiO₂-2 composite catalyst displays improved selectivity and yield rate toward NH₃ formation owing to enhanced suppressed competitive hydrogen evolution reaction and favorable NO₃⁻RR kinetics. Specifically, FePc/TiO₂-2 catalyst delivers an NH₃ yield rate of 17.4 mg h⁻¹ cm⁻², higher than 1.032675 mg h⁻¹ cm⁻² for Fe₂M MOFs catalyst in acid and that for most reported TiO₂-based catalysts in neutral/alkaline media.

3. Possible NO₃⁻RR mechanism of FePc/TiO₂ was proposed with the help of different N-containing intermediates detected by XPS and FTIR spectra. The energy-favorable pathway is proposed as *NO₃ → *NO₂ → *NO₂ → *NO → *NOH → *NH₂OH → *NH₃ and the step of *NO → *NOH is rate-determining.

4. An alkaline-acid hybrid Zn-NO₃⁻ battery (AAHZNB) for highly efficient NH₃ production was developed for the first time. Such AAHZNB based on FePc/TiO₂-2 cathode shows a high open-circuit potential of 1.99 V with an exceptional power density of 91.4 mW cm⁻², much higher than that for all the reported Zn-NO_x⁻ batteries.

5. The pollute-based AAHZNB can effectively power the environmental sulfur recovery with a current density of 35.6 mA cm⁻².

6. The N₂H₄ can replace the solid Zn anode and the newly developed all-pollutes-based N₂H₄-NO₃⁻ fuel cell enables the simultaneous N₂H₄/NO₃⁻ elimination, value-added NH₃ synthesis and electricity supply.

Please check the highlight on page 2 in the revised manuscript.

2) In line 37, where you discuss the Zn-NO₃⁻ batteries, please provide some literature review about the already existing ones and how this one holds more advantages than them.

Response: Thanks for your valuable suggestion. We are sorry that we can't discuss too

much in the abstract due to the page limitation. Here, we supplemented the literature review about the already existing Zn-NO₃⁻ batteries before the discussion of Fig. 4 in the manuscript.

Recently, many Zn-NO₃⁻ battery systems based on different catalyst cathodes have been reported (Energy Environ. Sci. 2021, 14, 3938; Adv. Energy Mater. 2022, 12, 2103872; Proc. Natl. Acad. Sci. U.S.A. 2023, 120, e2306461120; Adv. Mater. 2023. DOI: 10.1002/adma.202304508; Angew. Chem. Int. Ed. 2023, 135, e202305695; Energy Environ. Sci. 2023, 16, 2483). They can efficiently utilize the electrons generated in the NO₃⁻RR process, which can not only deliver a high theoretical energy density but also provide a feasible strategy for future NH₃ production and NO₃⁻-containing wastewater treatment. Zhi's group first developed a galvanic Zn-NO₃⁻ cell with an open circuit voltage (OCV) of 0.81 V and a power density of 0.87 mW cm⁻², based on a Pd/TiO₂ supported on carbon cloth as the cathode (Energy Environ. Sci. **2021**, 14, 3938). Subsequently, they demonstrated that such Zn-NO₃⁻ battery with OCV of 1.22 V can be rechargeable but irreversible, and oxygen evolution reaction occurs at the cathode during the charging process (Adv. Energy Mater. 2022, 12, 2103872). Interestingly, Jiang et al. recently proposed a rechargeable and reversible Zn-nitrogen flow battery and NH₃ can be oxidized to NO₂⁻ and further NO₃⁻ during the charging process. However, it lost the unique advantage of turning waste into treasure, and such a battery only exhibits an OCV of 1.39 V and a power density of 10.0 mW cm⁻² (Angew. Chem. Int. Ed. 2023, 135, e202305695). Up to now, the highest power density of Zn-NO₃⁻ battery has been reported by Zhou's group (Energy Environ. Sci. 2023, 16, 2991). Benefiting from the high NO₃⁻RR activity, the galvanic Zn-NO₃⁻ battery with OCV of 0.94 V shows a power density up to 70.7 mW cm⁻² with CuNi nanoparticles supported on Cu foil as the cathode in 3.5 M NaOH/44.3 g L⁻¹ NO₃⁻ (0.71 M NO₃⁻) catholyte.

Although great progress has been made recently, all reported Zn-NO₃⁻ batteries deliver limited voltages. The high power density and NH₃ yield/selectivity are still desired for this electrochemical cell system, which severely depends on the conditions of the cathodic part. For aforementioned excellent work about Zn-NO₃⁻ battery, they mainly aim to improve the catalytic current density by rational design NO₃⁻RR electrocatalysts. However, the overpotential for NO₃⁻RR of these electrocatalysts is very large, leading to low energy efficiency for NH₃ synthesis. As far as we know, all the reported Zn-NO₃⁻ batteries are equipped with NO₃⁻RR in neutral/alkaline conditions (Table R3). Although many reported open-circuit voltage (V) is larger than 1 V, the discharging onset voltages are lower than 1 V, resulting in limited power density. In this regard, an alkaline-acidic hybrid Zn-NO₃⁻ battery (AAHZNB) is highly attractive to offer a large power density with high NH₃ yield as acidic NO₃⁻RR can not only realize the large-current NH₃ synthesis but also harvest the neutralization energy of acid and alkali introduced by the pH difference between the two chambers separated by a bipolar membrane, which can efficiently avoid the direct neutralization of acid and alkali. As such, the developed AAHZNB in this work delivers a high open circuit voltage of 1.99 V and power density of 91.6 mW cm⁻², higher than all of the reported Zn-NO₃⁻ batteries up to now.

Table R3. Summary of the reported Zn-NO₃⁻ batteries. OCV is highlighted to show the highest value achieved by our alkaline-acid Zn-NO₃⁻ battery.

Catholyte	Anolyte	Cathode	OCV (V)	NH ₃ yield (mg h ⁻¹ cm ⁻²)	P (mW cm ⁻²)
0.1 M HNO₃ + 0.4 M KNO₃	6 M KOH	FePc/TiO₂-2	1.99	12.3	91.4
0.25 M LiNO ₃ + 5 M LiCl	5 M KOH	Pd/TiO ₂	0.81	0.54	0.87
0.2 M K ₂ SO ₄ + 0.05 M KNO ₃	1 M KOH	Fe/Ni ₂ P	1.22	0.38	3.25
1 M KOH + 0.05	5 M KOH	MP-Cu	1.27	1.292	7.56

M NO ₃ ⁻					
0.5 M K ₂ SO ₄ + 200 ppm NO ₃ ⁻	-	NiCu-SAA	1.51	2.1	12.7
1 M KOH + 1 M KNO ₃	1 M KOH + 0.02 M Zn(Ac) ₂	DM-Co	0.7	2.04	25.8
1 M KOH + 1 M NaNO ₃	1 M KOH	Ru-25CV/NF	1.2	2.9	51.5
0.1 M NaOH + 0.1 M NO ₃ ⁻	6 M KOH	NiCo ₂ O ₄	1.3	0.82	3.94
3 M KOH + 0.5 M NO ₃ ⁻	3 M KOH	Cu nanowire	0.943	2.125	14.1
0.1 M NaOH + 0.1 M NO ₃ ⁻	6 M KOH	Ir SAC Co ₃ O ₄	1.396	0.7633	5.6
-	0.1 M NaNO ₃	Fe ₂ TiO ₅	1.5	0.7803	5.6
1 M KOH + 0.1 M NO ₃ ⁻	6 M KOH	NiCoBDC@HsGDY	1.47	1.125	3.66
1 M KOH + 0.1 M NaNO ₃	1 M KOH	W-O-CoP	0.7	2.79	9.27
0.5 M Na ₂ SO ₄ + 0.1 M NaNO ₃	1 M KOH + 0.02 M Zn(Ac) ₂	RuFe NF	1.37	-	9.5
3 M KOH + 0.5 M NO ₃ ⁻	3 M KOH + 0.1M Zn(Ac) ₂	NiRu ball-flower	1.39	-	10
1 M KOH + 0.1 M KNO ₃	6 M KOH	Ru/β-Co(OH) ₂	1.48	6.46	29.87
3.5 M NaOH + 44.3 g L ⁻¹ NO ₃ ⁻	3.5 M NaOH	CuNi NPs/CF	0.94	18.1	70.7

Please check the highlighted part on page 19–22 in the revised manuscript and modified Table 2 in the revised Supplementary information.

3) In Fig. 2, please include ammonia current densities as well.

Response: Thank you for your insightful comments. As you suggested, we supplemented the NH₃ partial current densities (j_{NH_3}) of both TiO₂ and FePc/TiO₂-2, as shown in Fig. R21. Benefiting from the high NH₃ FE of FePc/TiO₂-2, its j_{NH_3} of FePc/TiO₂-2 is higher than that for TiO₂ at each potential. Specifically, FePc/TiO₂-2 could reach a j_{NH_3} of 219.7 mA cm⁻² at -0.65 V, much higher than that of TiO₂.

Fig. R21. Partial current densities of NH₃ (j_{NH_3}) of TiO₂ and FePc/TiO₂-2 for NO₃⁻RR in NO₃⁻ (pH 1).

Please check the highlighted text on page 12 in the revised manuscript and the newly added Fig. 18 in the revised Supplementary information.

4) NMR was done to confirm the presence of NH₃ for isotopic labelling cases. Please perform NMR for some of the N¹⁴ cases as well as UV-Vis. spectrometry is subjected to false positives especially for NH₃.

Response: Thanks for your valuable comment. As you suggested, we reperformed the NMR measurements for quantitative detection of NH₃ yield with ¹⁵NO₃⁻ and ¹⁴NO₃⁻ as the N sources. We first built the calibration curve using the standard (¹⁴NH₄)₂SO₄ and (¹⁵NH₄)₂SO₄, as displayed in Fig. R22. Then, we conducted the electrolysis at -0.45 V for in 0.5 M ¹⁵NO₃⁻ and ¹⁴NO₃⁻ electrolytes, respectively. The electrolytes were collected for further analysis. Typical double and triple peaks appear in NMR spectra for ¹⁵NH₄⁺ and ¹⁴NH₄⁺ (Fig. R23a), respectively, indicative that the nitrogen source of produced ammonia is NO₃⁻. Additionally, the NH₃ yield rate and FE obtained from the NMR method are finally recorded at 11.2 mg h⁻¹ cm⁻² and 85.0% with ¹⁴NO₃⁻ and 11.8 mg h⁻¹ cm⁻² and 88.7% with ¹⁵NO₃⁻, which are close to the UV-Vis. results (11.4 mg h⁻¹ cm⁻² and 88.6%), as shown in Fig. R23b, indicative of the reliability of the

experimental data for NH_3 determination.

Fig. R22. (a) NMR spectra of $(^{15}\text{NH}_4)_2\text{SO}_4$ standard solution with concentration from 100 ppm to 400 ppm. (b) Established calibration curve for $(^{15}\text{NH}_4)_2\text{SO}_4$ standard solution. (c) NMR spectra of $(^{14}\text{NH}_4)_2\text{SO}_4$ standard solution with concentration from 100 ppm to 400 ppm. (d) Established calibration curve for $(^{14}\text{NH}_4)_2\text{SO}_4$ standard solution.

Fig. R23. (a) The ^1H NMR spectra of standard $^{14}\text{NH}_4^+$ and $^{15}\text{NH}_4^+$, and obtained $^{14}\text{NH}_4^+$ and $^{15}\text{NH}_4^+$ from NO_3^- -RR with $^{14}\text{NO}_3^-$ and $^{15}\text{NO}_3^-$ as feed, respectively. (b) Comparison of NH_3 yield and NH_3 FE of FePc/ TiO_2 at -0.45 V obtained for NMR methods and UV-Vis. methods.

Please check the Fig. 3a, b, and highlighted parts on page 13 in the revised manuscript and the newly added Fig. 22, 23 in the revised Supplementary information.

5) In line 111 authors mention that TiO_2 enables more ammonia formation at lower pH. Please explain the mechanism and the process fundamentally in order to provide sufficient reasoning for this pH.

Response: Thanks for your comments. We are sorry for our oversight. Here, we would like to give a more detailed explanation from experimental and theoretical investigations to address your concerns.

First, the acidic media with abundant protons enables a more positive onset potential than that in neutral and alkaline media, which, however, require additional energy for water dissociation (ACS Catal. 2023, 13, 16, 10846–10854). Second, acidic media delivers a more rapid kinetics for NH_3 synthesis. Compared to neutral/alkaline media, acidic media shows a smaller Tafel slope and transfer efficiency (Fig. R24). In addition, the Nyquist plot for acidic medium displays a lower ohmic resistance loss because of higher ionic conductivity than neutral medium. These analyses contribute to the higher NH_3 partial current density at a given potential. Furthermore, we experimentally determined the activation energies (E_a) for NO_3^- -RR by analyzing the temperature-dependent kinetics of TiO_2 . Fig. R25 shows the LSV curves in 0.5 M NO_3^- with different pHs in the temperature range of 293 K to 303 K. By fitting the current densities measured at different temperatures using the Arrhenius equation (ACS Appl.

Mater. Interfaces 2023, 15, 18928–18939), E_a values can be obtained at different pH values. When the potential is -0.25 V, the E_a for pH 1 is 10.7 kJ mol^{-1} , much smaller than that for pH 7 and pH 13, suggesting a lower energy barrier for NO_3^- RR in acidic media.

In order to investigate the pH-dependent influences on the reaction pathways, we also examined the evolution of free energy plots at pH values of 1, 7, and 13. It is worth noting that the Gibbs free energy has been adjusted in accordance with pH corrections as detailed in a prior report (ACS Catal. 2021, 11, 14417–14427). Our analysis of the pH-dependent influences on the free energies of reaction intermediates is elucidated through the role of H^+ . Whether we consider the deoxygenation of $^*\text{NO}_x$ or the hydrogenation of $^*\text{NH}_y$, it becomes evident that NO_3^- RR exhibits favorable energetics when H^+ ions are readily available within an acidic medium. This trend is substantiated by the energy evolution diagram for NO_3^- RR presented in Fig. R26. As the pH increases to 7 and 13, a concomitant increase in the free energies for NO_3^- RR is observed due to the sluggish kinetics of H^+ produced from additional water dissociation. The foregoing analysis further suggests that an acidic environment may be more conducive to NO_3^- RR for TiO_2 .

Fig. R24. (a) Tafel plots and Nyquist plots of TiO_2 at different pH values.

Fig. R25. LSV curves of the TiO₂ catalysts at different temperatures in the electrolyte containing 0.5 M NO₃⁻ with (a) pH 1, (b) pH 7 and (c) pH 13. (d) An Arrhenius plot showing the linear relationship between logarithmic values of the reciprocal of the catalytic current densities and the reciprocal of absolute temperatures for NO₃⁻RR on the TiO₂ in 0.5 M NO₃⁻ electrolyte (pH 1) at -0.25 V.

Fig. R26. Gibbs free energies for NO₃⁻RR on TiO₂ at pH values of 1, 7 and 13.

Please check the modified Fig. 1c, the highlighted discussion on page 7–9 in the

revised manuscript and the newly added Fig. 1, 2, 10 in the revised Supplementary information.

6) In line 142, authors mention that they chose 0.5 M nitrate concentration for their studies. Please study the effect of nitrate concentration and include a figure pertaining to that.

Response: Thanks for your insightful comment. We are sorry for not completing the performance of FePc/TiO₂-2 at different concentrations. Following your suggestion, we performed the electrolysis at different potentials and different NO₃⁻ concentrations (0.1 M to 2 M). The obtained NH₃ yield and corresponding FE at different potentials and shown in Fig. R27. NH₃ formation rate of FePc/TiO₂-2 shows increased trends with increased NO₃⁻ concentrations from 0.1 M to 2 M in the solution at all potentials. The NH₃ formation rate reaches 6.84 mg h⁻¹ cm⁻² at -0.75 V in 0.1 M NO₃⁻, indicating that the FePc/TiO₂-2 are still active in low NO₃⁻ concentration. When the NO₃⁻ concentration is up to 2 M, the NH₃ formation rate still significantly increases and reaches 22.5 mg h⁻¹ cm⁻², suggesting that the FePc/TiO₂-2 can work under extreme environments. Besides, the calculated NH₃ FE also shows similar increased trends with increased NO₃⁻ concentration at each potential, suggesting that the high NO₃⁻ concentration fascinates the NH₃ formation. Specifically, the maximum NH₃ FE values for 0.1 M, 0.2 M, 0.3 M 0.4 M and 0.5 M NO₃⁻ electrolytes using FePc/TiO₂-2 are determined as 77.0%,79.2%, 83.2%, 87.9% and 90.6%, respectively. The NH₃ FE in 2 M NO₃⁻ is 92.7%, which is close to that in 0.5 M NO₃⁻ electrolytes. Therefore, we chose 0.5 M for NO₃⁻RR in the next explorations.

Fig. R27. (a) NH₃ yield and (b) NH₃ FE of FePc/TiO₂-2 at different potentials and different NO₃⁻ concentrations of 0.1 M, 0.2 M, 0.3 M, 0.4 M, 0.5 M and 2 M.

Please check the highlighted discussion on page 11 in the revised manuscript and the newly added Fig. 15, 16 in the revised Supplementary information.

7) In line 154, authors claim that hydrazine wasn't formed. Please explain how hydrazine was quantified in this study.

Response: Thanks for your kind reminder. Hydrazine (N₂H₄) in the electrolytes was detected by the Watt-Chrisp method (Anal. Chem. 1952, 24, 2006–2008). In detail, a mixture of ethanol (100 mL), para(dimethylamino) benzaldehyde (2.0 g) and HCl (concentrated, 12 mL) were used as a color reagent. Then, 2 mL color reagent was added into 2 mL electrolyte. After 30 min, the absorbance was measured at a wavelength of 458 nm. The concentration-absorbance calibration curve was built using a series of concentrations known as standard N₂H₄ solutions dissolved in the electrolyte (Fig. R28a,b). It should be noted that the UV-Vis. adsorption curves of the electrolytes

collected at different potentials are totally overlapped with the curve for a standard solution with N_2H_4 concentration of 0 (Fig. R28c,d), indicating that almost no hydrazine was formed during NO_3^- RR for both FePc/TiO₂-2 and TiO₂ catalysts.

Fig. R28. (a) UV-Vis. adsorption spectra. (b) standard curves for N_2H_4 at pH 1. (c) UV-Vis. adsorption curves for (a) TiO_2 and (b) $\text{FePc/TiO}_2\text{-2}$ at pH 1 with 0.5 M NO_3^- after electrolysis at different potentials for 30 min.

Please check the highlighted discussion on page 12 in the revised manuscript and Fig. 7, 20 in the revised Supplementary information.

REVIEWERS' COMMENTS

Reviewer #1 (Remarks to the Author):

My technical concerns raised in the previous review report have been mostly addressed in the revised manuscript.

Reviewer #2 (Remarks to the Author):

This work demonstrates the attractive potential of acidic NO₃-RR for NH₃ electrosynthesis, develops a high-performance hybrid Zn-NO₃- battery, and further broadens the field of fuel cells by using NO₃- as fuel. These systematic findings provide valuable information for the reasonable development of suitable host materials with enhanced Zn-NO₃- battery. Considering the solidness of the manuscript including synthesis, materials characterization, and electrochemical studies, I recommend to publish the work in Nature Communications.

Reviewer #3 (Remarks to the Author):

The authors have addressed concerns.

Response to Reviewers

Dear Reviewers:

We thank you for your comments concerning our manuscript entitled “Electrochemical Nitrate Reduction in Acid Enables High-Efficiency Ammonia Synthesis and High-Voltage Pollutes-Based Fuel Cells”. We provide a point-by-point response below.

Reviewer #1 (Remarks to the Author):

My technical concerns raised in the previous review report have been mostly addressed in the revised manuscript.

Response: We deeply thank the reviewer’s comments.

Reviewer #2 (Remarks to the Author):

This work demonstrates the attractive potential of acidic NO₃-RR for NH₃ electrosynthesis, develops a high-performance hybrid Zn-NO₃- battery, and further broadens the field of fuel cells by using NO₃- as fuel. These systematic findings provide valuable information for the reasonable development of suitable host materials with enhanced Zn-NO₃- battery. Considering the solidness of the manuscript including synthesis, materials characterization, and electrochemical studies, I recommend to publish the work in Nature Communications.

Response: We deeply thank the reviewer’s comments.

Reviewer #3 (Remarks to the Author):

The authors have addressed concerns.

Response: We deeply thank the reviewer’s comments.